# A lattice grain model of hillslope evolution

Gregory E. Tucker[1], Scott W. McCoy[2], and Daniel E.J. Hobley[3]

[1]Cooperative Institute for Research in Environmental Science (CIRES) and Department of Geological Sciences, University of Colorado, Boulder, CO 80305 USA
[2]Department of Geological Sciences and Engineering, University of Nevada, Reno, NV, 89557 USA
[3]School of Earth and Ocean Sciences, Cardiff University, Cardiff CF10 3AT Wales

*Correspondence to:* Greg Tucker (gtucker@colorado.edu)

**Abstract.** This paper describes and explores a new continuous-time stochastic cellular automaton model of hillslope evolution. The Grain Hill model provides a computational framework with which to study slope forms that arise from stochastic disturbance and rock weathering events. The model operates on a hexagonal lattice, with cell states representing fluid, rock, and grain aggregates that are either stationary or in a state of motion in one of the six cardinal lattice directions. Cells representing near-surface soil material undergo stochastic disturbance events, in which initially stationary material is put into motion. Net downslope transport emerges from the greater likelihood for disturbed material to move downhill than to move uphill. Cells representing rock undergo stochastic weathering events in which the rock is converted into regolith. The model can reproduce a range of common slope forms, from fully soil mantled to rocky or partially mantled, and from convex-upward to planar shapes. An optional additional state represents large blocks that cannot be displaced upward by disturbance events. With the addition of this state, the model captures the morphology of hogbacks, scarps, and similar features. In its simplest form, the model has only three process parameters, which represent disturbance frequency, characteristic disturbance depth, and baselevel lowering rate, respectively. Incorporating physical weathering of rock adds one additional parameter, representing the characteristic rock weathering rate. These parameters are not arbitrary but rather have a direct link with corresponding parameters in continuum theory. Comparison between observed and modeled slope forms demonstrates that the model can reproduce both the shape and scale of real hillslope profiles. Model experiments highlight the importance of regolith cover fraction in governing both the downslope mass transport rate and the rate of physical weathering. Equilibrium rocky hillslope profiles are possible even when the rate of baselevel lowering exceeds the nominal bare-rock weathering rate, because increases in both slope gradient and roughness can allow for rock weathering rates that are greater than the flat-surface maximum. Examples of transient relaxation of steep, rocky slopes predict the formation of a regolith-mantled pediment that migrates headward through time while maintaining a sharp slope break.

## 1 Introduction

Hillslopes take on a rich variety of forms. Their profile shapes may be convex-upward, concave-upward, planar, or some combination of these. Some slopes are completely mantled with soil, whereas others are bare rock, and still others draped

in a discontinuous layer of mobile regolith. The processes understood to be responsible for shaping them are equally varied, ranging from disturbance-driven creep to dissolution to large-scale mass movement events.

Considerable research has been devoted to understanding the evolution of soil-mantled slopes that are primarily governed by disturbance-driven creep, such as down-slope soil transport by biotic and abiotic soil-mixing processes. As a result, the geomorphology community has mathematical models that account well for observed slope forms and patterns of regolith thickness (e.g., Roering, 2008). Furthermore, stochastic-transport theory provides a mechanistic link between the statistics of particle motion, the resultant average rates of downslope transport, and the emergence of convex-upward, soil-mantled slope forms (Culling, 1963; Roering, 2004; Foufoula-Georgiou et al., 2010; Furbish et al., 2009; Furbish and Haff, 2010; Tucker and Bradley, 2010).

One gap that remains, however, lies in understanding steep, rocky slopes (Figure 1). "Rocky" implies slopes that lack a continuous soil cover (e.g., Howard and Selby, 1994, and references therein); here, transport laws that assume the existence of such a cover no longer apply. "Steep" implies angles approaching or exceeding the effective angle of repose for loose, granular material, so that ravel may be an important transport mode (e.g., Gabet, 2003; Roering and Gerber, 2005; Lamb et al., 2011; Gabet and Mendoza, 2012) and particles have the potential to fall as soon as they are released from bedrock. This type of relatively fast, long-distance transport does not fit comfortably in the framework of standard diffusion-based models of hillslope soil transport, which derive from an underlying assumption that the characteristic length scale of motion is short relative to the length of the slope.

Rocky slopes are rarely completely barren. More commonly, they have a patchy cover of loose material, which may either retard rock weathering by shielding the rock surface from moisture or temperature fluctuations, or enhance it by trapping water and allowing limited plant growth. A discontinuous cover does not fit easily within the popular exponential-decay regolith-production models (e.g., Heimsath et al., 2012; Lamb et al., 2013), which assume an essentially continuous soil mantle.

An additional issue, which pertains to both rocky and soil-mantled slopes, is the connection between sediment movement at the scale of individual "motion events," and the resulting longer-term average sediment flux, which forms the basis for continuum models of hillslope evolution. Recent theoretical and experiment work has begun to forge a mechanistic connection between these scales (Culling, 1963, 1965; Furbish et al., 2009; Furbish and Haff, 2010; Tucker and Bradley, 2010; Gabet and Mendoza, 2012; Lamb et al., 2013). However, the community's resources for computational analysis of particle-level dynamics remain limited, lagging behind developments in understanding sediment transport in coastal environments (Drake and Calantoni, 2001) and rivers (McEwan and Heald, 2001; MacVicar et al., 2006; Furbish and Schmeeckle, 2013; Schmeeckle, 2014).

To further our understanding of how grain-level weathering and transport processes translate into hillslope evolution, both for hillslopes in general and rocky slopes in particular, it would be useful to have a computational framework with which to conduct experiments. Ideally, such a framework should be sophisticated enough to capture the essence of weathering and granular mechanics, while remaining simple enough to involve only a small number of parameters and provide reasonable computational efficiency.

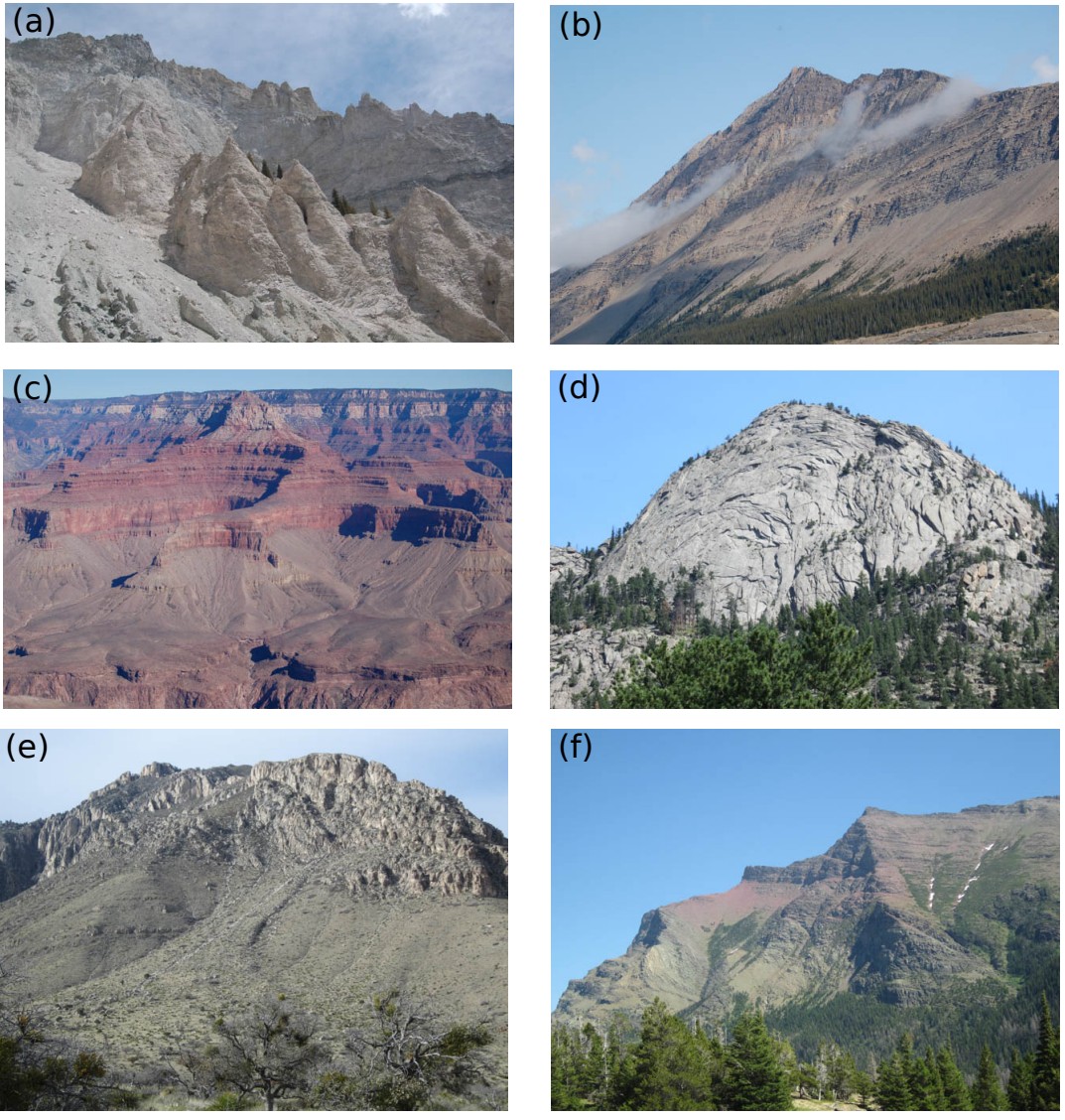

**Figure 1.** Examples of rocky hillslopes, sometimes referred to as Richter slopes. (a) Chalk Cliffs, Colorado, USA. (b) Canadian Rockies. (c) Grand Canyon, Arizona, USA. (d) Rocky Mountain National Park, Colorado, USA. (e) Guadeloupe Mountains, Texas, USA. (f) Waterton Lakes National Park, Canada (photos by G.E. Tucker).

Our aim in this paper is to describe one such computational framework, test whether it is capable of reproducing commonly observed hillslope-profile forms, and examine how its parameters relate to the bulk-behavior parameters used in conventional continuum models of soil creep and regolith production. The model uses a pairwise, continuous-time stochastic (CTS) approach to combine a Lattice Grain model with rules for stochastic bedrock-to-regolith conversion ("weathering") and disturbance of

surface regolith particles. One goal of this event-based approach is to study how bulk behavior, such as the diffusion-like net downslope transport of soil, can emerge from a large ensemble of stochastic events. In this paper, we present the "Grain Hill" model, and examine its ability to reproduce three common types of slope profile: (1) convex-upward, soil-mantled slopes (Figure 2a,b), (2) quasi-planar rocky slopes (Figure 2c,d), and (3) cliff-rampart morphology in layered strata (Figure 2e,f).

We begin with a description of the modeling technique. We then present results that illustrate the macroscopic behavior of the model under a variety of boundary conditions, and define the relationship between the cellular model's parameters and the parameters of conventional continuum mechanics models for hillslope evolution.

## 2    Model Description

The model combines a cellular automaton representation of granular mechanics with rules for weathering of rock to regolith
and for episodic disturbance of regolith. Cellular automata are widely used in the granular mechanics community, because they can represent the essential physics of granular materials at a reasonably low computational cost. Because the principles are often similar to those of lattice-gas automata in fluid dynamics (e.g., Chen and Doolen, 1998), cellular automata for granular mechanics are sometimes referred to as Lattice Grain models (LGrMs) (Gutt and Haff, 1990; Peng and Herrmann, 1994; Alonso and Herrmann, 1996; Károlyi et al., 1998; Károlyi and Kertész, 1999, 1998; Martinez and Masson, 1998; Désérable,
2002; Cottenceau and Désérable, 2010; Désérable et al., 2011).

### 2.1    CTS Lattice Grain Model

Our approach starts with a two-dimensional (2D) continuous-time stochastic ("CTS") Lattice Grain cellular automaton. A cellular automaton can be broadly defined as a computational model that consists of a lattice of cells, with each cell taking on one of $N$ discrete states (represented by an integer value). These states evolve over time according to a set of rules that describe
transitions from one state to another as a function of a particular cell's immediate neighborhood. A continuous-time stochastic model is one in which the timing of transitions is probabilistic rather than deterministic. Whereas transitions in traditional cellular automata occur in discrete time steps, in a CTS model they are both stochastic and asynchronous. A CTS model can be viewed as a type of Boolean Delay Equation (Ghil et al., 2008), though the number of possible states is not necessarily limited to just two.

The method we present here, which we will refer to as the Grain Hill model, is implemented in the Landlab modeling framework (Hobley et al., 2017). The Lattice Grain component, on which Grain Hill builds, is described in detail by Tucker et al. (2016). Here we present only a brief overview of the Lattice Grain model's rules and behavior. The framework is based on the pairwise ("doublet") method developed by Narteau and colleagues (Rozier and Narteau, 2014), which has been applied to problems as diverse as eolian dune dynamics (Narteau et al., 2009; Zhang et al., 2010, 2012) and the core-mantle interface
(Narteau et al., 2001).

In the basic CTS Lattice Grain model, the domain consists of a lattice of hexagonal cells. Each cell is assigned one of eight states (Table 1, states 0–7). These states represents the nature and motion status of the material: state 0 represents fluid (an

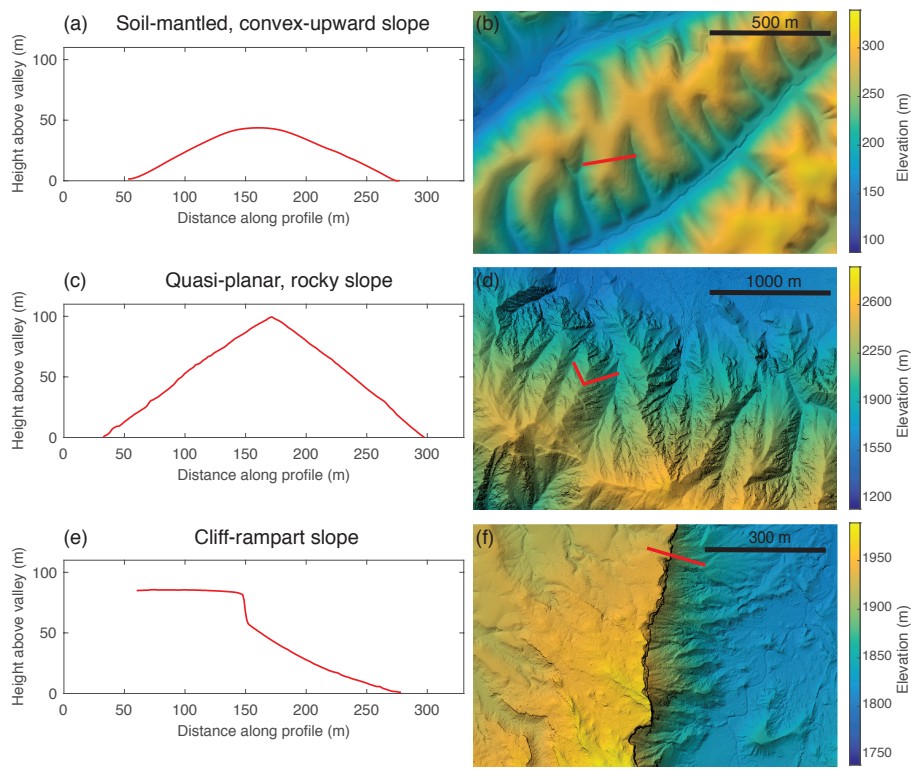

**Figure 2.** Examples of three characteristic types of hillslope profile. Red line in mapview depicts hillslope profile location. (a, b) Soil-mantled, convex-upward slope (Gabilan Mesa, California, USA). (c, d) Quasi-planar, thinly mantled slope (Yucaipa Ridge, California, USA). (e, f) Cliff formed in resistant Tertiary laccolithic intrusive rocks overlying Jurassic sedimentary rocks (Cedar Mountain, Utah, USA).

**Table 1.** States in the Grain Hill model.

| State | Description |
| --- | --- |
| 0 | Fluid |
| 1 | Grain moving upward |
| 2 | Grain moving up and right |
| 3 | Grain moving down and right |
| 4 | Grain moving down |
| 5 | Grain moving down and left |
| 6 | Grain moving up and left |
| 7 | Resting grain |
| 8 | Rock |
| (9) | Block (optional) |

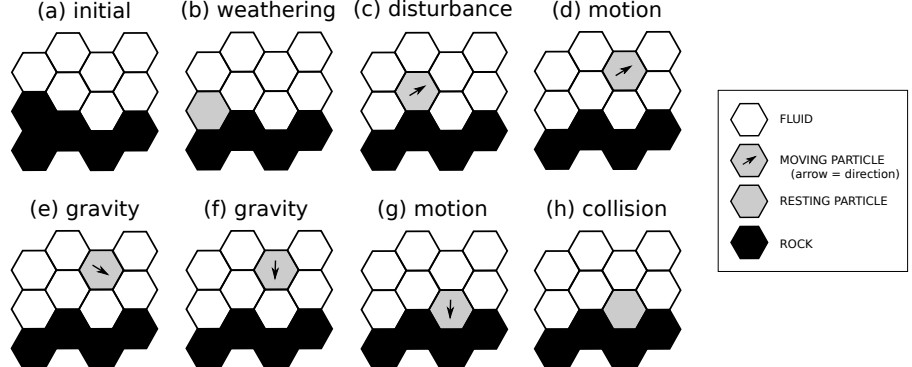

**Figure 3.** Hypothetical time sequence of transition events, from upper left to lower right, illustrating several of the states and transitions in the Grain Hill model. Note that although this example shows a single particle in motion, it is possible for multiple cells to exist in a state of motion at any given time.

"empty" cell into which a solid particle can move), states 1–6 represent a grain moving in one of the six lattice directions, and state 7 indicates a stationary grain (or aggregate of grains, as discussed below). For purposes of modeling hillslope evolution, we add an additional state (8) to represent rock, which is immobile until converted to granular material, representing regolith. An optional additional state (9) is used to model large blocks, as described below. Figure 3 shows several of these states in the form of a time sequence of transition events. Note that the timing of transition events is purely stochastic; there are no time steps in the usual sense.

Like other Lattice Grain models, the CTS Lattice Grain model is designed to represent, in a simple way, the motion and interaction of an ensemble of grains in a gravitational field. The physics of the material are represented by a set of transition

rules, in which a given adjacent pair of states is assigned a certain probability per unit time of undergoing a transition to a different pair. For example, consider a vertically aligned pair of cells in which the top cell has state 4 (moving downward) and the bottom cell state 0 (empty/fluid) (Figure 3f). Downward motion (falling) is represented by a transition in which the two states switch places (Figure 3g).

The stochastic pairwise transitions in the CTS Lattice Grain model are treated as Poisson processes. The probability density function for the waiting time, $t$, to the next transition event at a particular pair is given by an exponential function with a rate parameter $r$, which has dimensions of inverse time:

$$p(t) = re^{-rt}. \tag{1}$$

Each transition type is associated with a rate parameter that represents the speed of whichever process the transition is designed
to represent. To implement these transitions, the CTS Lattice Grain model steps from one transition to the next, rather than iterating through time steps of fixed duration. Whenever the state of one or both cells in a particular pair changes, if the new pair is subject to a transition, the time at which the transition is scheduled to occur is added to a queue of pending events. The soonest among all pending events is chosen for processing, and the process repeats until either the desired run time has completed or there are no further events in the queue. Further details on the implementation and algorithms are provided in
Tucker et al. (2016).

Grain motion through fluid is represented by a transition involving a moving grain and an adjacent fluid cell in the direction of the grain's motion: the two cells exchange states, representing the motion of the grain into the fluid-filled cell, and the replacement of the grain's former location with fluid (Figure 3, c-d and f-g). During this transition, the grain's motion direction remains unchanged (Figure 4, top left). Note that the lattice itself never moves; rather, material motion is represented simply
by an exchange of grain and fluid states between an adjacent pair of cells.

Gravity is represented by transitions in which a rising grain decelerates to become stationary, a stationary grain accelerates downward to become a falling particle, and a grain moving upward at an angle accelerates downward to move downward at an angle (Figure 5). An additional rule allows for acceleration of a particle resting on a slope: a stationary particle adjacent to a fluid cell below it and to one side may transition to a moving particle (Figure 5, bottom row). Importantly for our purposes,
this latter rule effectively imposes an angle of repose at $30°$.

For gravitational transitions, the rate parameter, $r_g$, is determined by considering the time it would take for an initially stationary object to fall a distance of one cell width under gravitational acceleration without fluid drag. This works out to be

$$r_g = \sqrt{2\delta/g}, \tag{2}$$

where $\delta$ is cell width and $g$ is gravitational acceleration. This rate parameter is used for all of the gravitational transitions
illustrated in Figure 5.

Because of the stochastic treatment of all transitions—including gravitational ones—it is possible for grains in the model to hover in mid-air for a brief period of time before plunging downward (e.g., Coyote, 1949). For purposes of modeling hillslope evolution, this is fine; what matters most is that there is a distinct time-scale gap between "fast" (large rate constant) processes

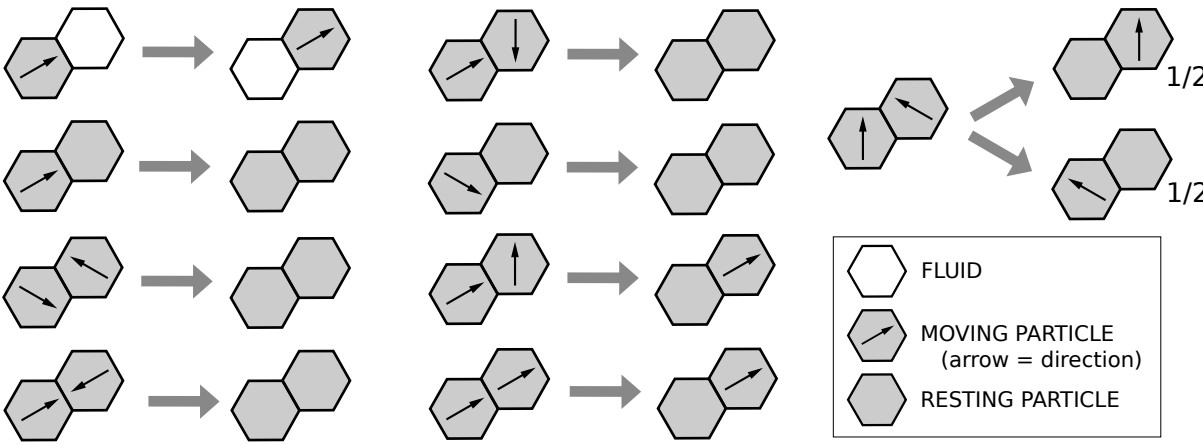

**Figure 4.** Rules for motion and frictional (inelastic) collisions, illustrated here for one of the six lattice directions.

associated with grain motion, and "slow" (small rate constant) processes associated with weathering and disturbance, which are described below. First, however, we must consider frictional interactions among moving particles.

We assume that biophysical disturbance events such as the growth of roots and burrowing by animals, and the settling motions that follow, tend to impart low kinetic energy, with "low" defined as ballistic displacement lengths that are short
relative to hillslope length and comparable to or less than the characteristic disturbance-zone thickness. We consider such motions to be dominated by frictional dissipation rather than by transfer of kinetic energy by elastic impacts. This view is similar to the reasoning of Furbish et al. (2009) that the mean-free-path of mobile grains will typically be short relative to hillslope length, scaling with the grain radius and particle concentration. For this reason, unlike the original Lattice Grain model of Tucker et al. (2016), the present formulation includes only inelastic collisions (Figure 4). These inelastic (frictional)
collisions are represented by a set of rules in which one or both colliding particles become stationary, representing loss of momentum and kinetic energy as a result of the collision. The particular choices for frictional interaction are motivated simply by the geometry of the problem. They are non-unique in the sense that one could imagine reasonable alternatives to the rules illustrated in Figure 4; however, the details of frictional interactions have little influence on the outcomes of the Grain Hill model. In the general Lattice Grain CTS model, the rate parameter for frictional transitions is set equal to the product of the
gravitational parameter and a dimensionless friction factor, $f \in [0, 1]$ (there is also a corresponding elastic factor equal to $1 - f$). In the Grain Hill implementation explored in this paper, $f = 1$, such that particle collisions are purely frictional.

One limitation of the CTS Lattice Grain model is that falling grains do not accelerate through time; instead, they have a fixed transition probability that implies a statistically uniform downward fall velocity. This treatment is obviously unrealistic for particles falling in a vacuum, though it is consistent with a terminal settling velocity for grains immersed in fluid. Consistent
with the above reasoning, the relatively short ballistic displacement lengths asserted for the modeled hillslopes also reduce the importance of this assumption, as a particle would typically have little time to accelerate before impacting another particle.

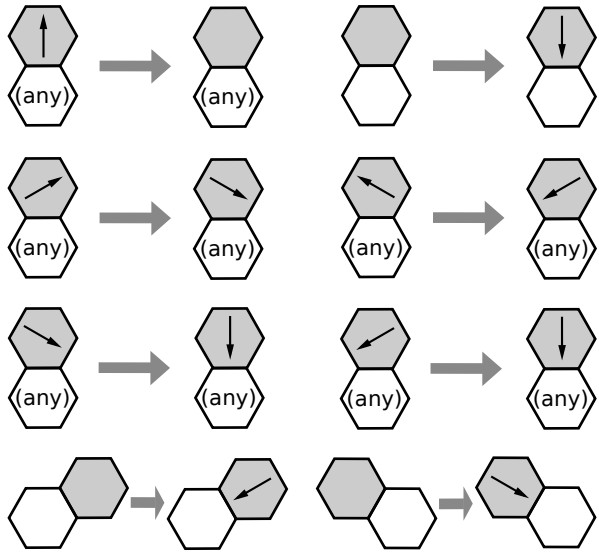

**Figure 5.** Illustration of gravitational rules. The bottom row shows the "falling on a slope" rule, which effectively imposes a 30° angle of repose. Modified from Tucker et al. (2016).

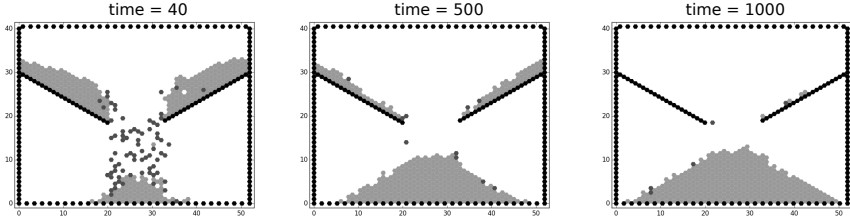

**Figure 6.** Lattice Grain simulation of emptying of a silo. Light-shaded grains are stationary; darker-shaded ones are in motion. Black cells are walls (rock). From Tucker et al. (2016).

Tests of the CTS Lattice Grain model show that it reproduces several basic aspects of granular behavior (Tucker et al., 2016). For example, when gravity and friction are deactivated, the model conserves kinetic energy. When gravity and friction are active, the model reproduces some of the common behaviors observed with granular materials. For example, Figure 6 illustrates a simulation of the emptying of a silo to form an angle-of-repose grain pile. For our purposes, what matters most is simply that the model captures, in a reasonable way, the response of particles on a slope to episodic disturbance events.

## 2.2 Weathering and Soil Creep

Weathering of rock to form mobile regolith is modeled with a transition rule: when a rock cell lies adjacent to a fluid cell (which here is assumed to be air), there is a specified probability per unit time, $w$ [1/T], of transition to a grain-air pair (Figure 3a-b,

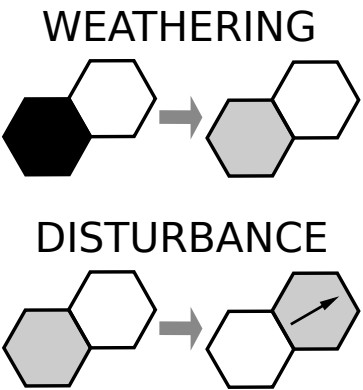

**Figure 7.** Transitions representing rock-to-regolith transformation by weathering (top), and regolith disturbance (bottom), in which a stationary particle becomes mobile and switches position with a air cell. Illustration represents one of the six possible orientations.

and Figure 7, top). In other words, $w$ is the Poisson rate constant for the weathering transition process. This treatment means that the effective maximum expected weathering rate, in terms of the propagation of a weathering front, is cell diameter, $\delta$, times $w$. An indirect consequence of this approach is that the weathering rate declines with increasing regolith thickness. As average regolith thickness increases, the fraction of the surface where rock is in contact with air diminishes, and consequently so too does the average transition rate. A limitation of the approach is that when the rock is completely mantled, no further weathering can take place. We explore the consequences of this rule below, and compare it with the behavior of continuum regolith-production models.

Soil creep is modeled by a transition rule that mimics the process of episodic disturbance of the mobile regolith (which we use here as a generic term that includes various forms of unconsolidated granular material, such as soil, colluvium, and scree). For each resting grain that is adjacent to an air cell, there is a specified probability per unit time, $d$ [1/T], that the regolith and air will exchange places, representing movement (Figure 3b-c). The regolith cell is also converted from a stationary state to a state of motion (Figure 7, bottom). An advantage of this approach is that it mimics, in a general way, the effectively stochastic disturbance processes that are understood to drive soil creep.

Our definition of $d$ is closely related to the activation rate, $N_a$, in the probabilistic theory for soil creep developed by Furbish et al. (2009). When combined with the Lattice Grain gravitational rules, the resulting cellular model captures both the scattering (disturbance) and settling (gravitational) behavior articulated by Furbish et al. (2009). In the Grain Hill cellular model, as in their theory, downslope regolith flux arises because, on average, scattering occurs normal to the local surface while setting is vertical. The Grain Hill model includes an additional element not present in the Furbish et al. (2009) theory: an increase in (downward) scattering distance for particles on slopes steeper than $30°$. This behavior, as illustrated below, promotes a nonlinear relationship between gradient and flux, and leads to the possibility of threshold slopes.

Note that the weathering and disturbance rate constants ($w$ and $d$, respectively) are understood to be considerably smaller than the gravitational rate constant, $r_g$. As noted above, a key concept here is that there are two distinct time scales: a short time scale associated with grain motion, and a much longer scale associated with weathering and disturbance frequency.

## 2.3 Cells as Grain Aggregates

Natural regolith disturbance events usually impact many grains at once. Raindrop impacts on bare sediment typically dislodge several grains at once (Furbish et al., 2007). Excavation of an animal burrow disturbs a volume a grains equal to the volume of the burrow, and the fall of a tree mobilizes a volume of regolith similar to the volume of the tree's root mound. Observations of such processes suggest that there may be a characteristic volume of disturbance that in some cases may be much larger than the volume of a single grain. For this reason, we envision regolith cells as being grain aggregates, with a length scale (width of

a cell) $\delta$ and a volume scale $\delta^3$.

## 2.4 Initial and Boundary Conditions

The 2D model domain represents the cross-section of a hypothetical hillslope, on which particles move within the cross-sectional plane. Any regolith cells that reach the model's side or top boundaries disappear. This treatment is meant to represent the presence of a stream channel at the base of each side of the model hillslope; particles reaching these channels are assumed

to be eroded. Progressive lowering of baselevel at the two model boundaries is treated by moving the interior cells upward away from the lower boundary, and adding a new row of rock or regolith cells along the bottom row. A new row of cells is added at time intervals of $\tau$.

Cells around the lattice perimeter retain their initial states. If, for example, a transition occurs in which a grain "moves" into a fluid cell on the lattice perimeter, its former location will correctly transition to a fluid cell, but the perimeter cell itself

will retain its status as a fluid cell. Effectively, this treatment means that grains or blocks reaching either of the two vertical boundaries are instantly eroded.

The initial condition for most runs presented here has the bottom two rows filled with regolith grains. The lower left and lower right cells are assigned to be rock, which represents the baselevel (and incidentally helps keep a consistent color scheme among different model configurations, because the rock state is always present). The rest of the domain is initialized as air

cells.

## 2.5 Scaling and Nondimensionalization

The basic model has four parameters: the disturbance rate, $d$ [cells/T], weathering rate, $w$ [cells/T], baselevel lowering interval, $\tau$ [T], and width of domain, $\lambda$ [cells]. The baselevel lowering timescale $\tau$ represents the time interval between episodes of relative uplift in which the interior domain is lifted by one cell relative to its side boundaries. The domain width might properly

be considered a boundary condition rather than a parameter, but we include it here with an eye toward examining how slope width impacts hillslope properties such as mean height. Once we define the width of a cell, $\delta$ [L], we can define versions of

these four parameters that explicitly incorporate this length scale:

$$D = d\delta, \tag{3}$$

$$W = w\delta, \tag{4}$$

$$U = \delta/\tau, \tag{5}$$

$$L = \lambda\delta. \tag{6}$$

Consider the case of dynamic equilibrium, in which the rate of baselevel lowering is balanced by the hillslope's rate of erosion. The mean height of this steady state hillslope, $H$, is a function of the above four parameters plus the characteristic length scale $\delta$, such that we end up with a total of six variables:

$$H = f(D, W, U, L, \delta). \tag{7}$$

Buckingham's Pi Theorem dictates that these six variables, which collectively include dimensions of length and time, may be grouped into four dimensionless quantities:

$$\frac{H}{\delta} = f\left(\frac{D}{U}, \frac{W}{U}, \frac{L}{\delta}\right) \tag{8}$$

The ratio $d' = d\tau = D/U$ is a dimensionless disturbance rate. Similarly, $w' = w\tau = W/U$ is a dimensionless weathering rate. Noting the definitions above, equation (8) is equivalent to

$$h = f(d', w', \lambda), \tag{9}$$

where $h = H/\delta$ is dimensionless hillslope height. Hence, we have a dimensionless property of the hillslope, $h$, that depends uniquely on three other non-dimensional variables, representing disturbance rate, weathering rate, and length.

One can similarly define a dimensionless regolith thickness, $r = R/\delta$, where $R$ is the dimensional equivalent; it too should depend on the three dimensionless parameters that represent disturbance rate $d'$, weathering rate, $w'$, and hillslope length, $\lambda$, respectively. For a hillslope composed entirely of regolith, $r$ and $h$ depend solely on $d'$ and $\lambda$. Finally, we define a fractional regolith cover $F_r$. In the Grain Hill model, $F_r$ is calculated as the number of air-regolith cell pairs divided by the total number of cell pairs that juxtapose air with either regolith or rock.

## 2.6 Blocks

The foregoing model is designed to represent regolith as grain aggregates composed of gravel-sized and finer grains: material fine enough that it is susceptible to being moved by processes such as animal burrowing, frost heave, tree throw, and so on. Some hillslopes, however, are adorned with grains that are simply too large to be displaced significantly by such processes. For example, Glade et al. (2017) presented a case study and model of slopes formed beneath a resistant rock unit that periodically sheds meter-scale or larger blocks. On at least some of these types of slope, the distance between surface blocks and their source unit is considerably greater than the distance that they could roll during an initial release event (Duszyński and Migoń,

2015). This observation implies that the blocks are transported down slope by a process of repeated undermining. Glade et al. (2017) hypothesized that erosion of soil beneath and immediately downhill can cause a block to topple, and hence move a distance comparable to its own diameter in each such event.

We wish to capture this form of "too big to disturb" behavior in the Grain Hill model. The CTS approach, at least as it is defined here, does not lend itself to variations in grain size or geometry. Instead, we introduce an additional type of particle that represents the behavior of blocks rather than treating their difference in size explicitly. In a sense, the approach can be viewed as treating blocks as having greater density, rather than greater size, than other grains. A block particle differs from a normal regolith cell in that it cannot be scattered upward by disturbance. Motion of a block particle can only occur under two circumstances: when it lies directly above an air cell (in which case it falls vertically, trading places with the air cell), and when it lies above and to the side of an air cell (in which case it falls downslope at a $30°$ angle, with probability per time $d$). These rules mimic the undermining process discussed by Glade et al. (2017).

As in the Glade et al. (2017) model, block particles can also undergo weathering. Here, weathering is again treated in a probabilistic fashion: blocks form from weathering of bedrock, at probability per time $w$. Once created, a block can undergo a conversion to normal regolith with probability $w$ when it sits adjacent to an air cell. This treatment of blocks captures, in a simple way, the weathering of blocks as they move down slope. For purposes of this paper, the block component is included simply to test whether a cellular automaton treatment produces results that are qualitatively consistent with observations, and also consistent with the hybrid continuum-discrete model of Glade et al. (2017) and Glade and Anderson (2017).

## 3   Results

### 3.1   Fully soil-mantled hillslope

We start by considering the case of fully soil-mantled hillslopes, in which the supply of mobile regolith is effectively unlimited (Figure 2a,b). Under this condition, the Grain Hill model represents a testable mechanistic hypothesis: that a transport-limited, soil-mantled hillslope behaves essentially as a granular medium subject to periodic, quasi-random disturbance events. This concept was also the essence of the acoustic-disturbance experiments by Roering et al. (2001). To test the hypothesis, we run the Grain Hill model with a constant rate of material uplift relative to baselevel until the system reaches quasi-steady state, to determine whether its steady form is smoothly convex upward (when the gradient is below the failure threshold) to planar (when the gradient lies at or near the failure threshold). Model runs were performed using a 251-row by 580-column lattice. Disturbance rates were varied from 0.001 yr$^{-1}$ to 0.1 yr$^{-1}$ and intervals between relative-uplift events from 100 to 10,000 yr.

Results show that the Grain Hill model produces parabolic to planar hillslope forms, depending on the ratio of disturbance to uplift rates, which is encapsulated in the dimensionless parameter $d'$ (Figure 8). At high $d'$ (frequent disturbance and/or slow baselevel fall), hillslope relief is low and the form is smoothly convex upward (Figure 8, lower right panels). At somewhat lower $d'$, the lower part of the slope approaches a threshold angle while the upper part remains smoothly convex (Figure 8, middle diagonal panels). At low $d'$, the form becomes predominantly planar and achieves a threshold relief that is insensitive for further increases in $d'$ (Figure 8, upper left panels).

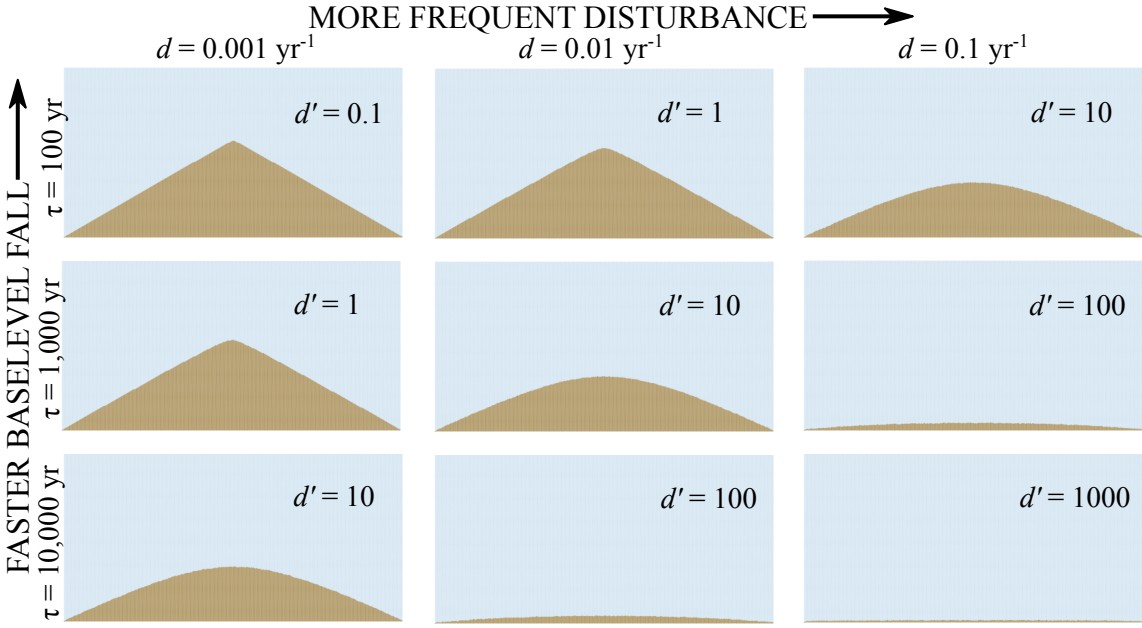

**Figure 8.** Equilibrium topographic cross-sections using only regolith particles (no rock) and a variety of disturbance frequencies ($d$) and time interval between baselevel fall events ($\tau$). Fast basal incision and/or infrequent disturbance leads to planar threshold hillslopes; slow basal incision and/or frequent disturbance leads to parabolic hillslopes.

Scaling of mean height as a function of $d'$ is shown in Figure 9. The figure shows results for 125 model runs spanning two orders of magnitude in each parameter ($d$, $\tau$, and $\lambda$) in half-decade intervals. The 125 runs represent a $5 \times 5 \times 5$ grid of experiments, in which each grid point represents a particular combination of the three parameters $d$, $\tau$, and $\lambda$.

For any given hillslope length, there are three regimes of behavior. Low $d'$ (upper left of graph) leads to threshold hillslopes, in which relief depends only on hillslope length. Under moderate $d'$, mean height scales inversely with $d'$, as expected from linear diffusion theory. At high $d'$, we have a finite-size regime in which dimensionless hillslope mean height is comparable to the disturbance scale, $\delta$ (cell size in the model); in other words, the hill is only one or a few cells high.

The behavior of the Grain Hill model in its simple, transport-limited configuration can be compared to diffusion theory, which relates volumetric sediment flux per unit contour length, $\mathbf{q}_s$, to topographic gradient:

$$\mathbf{q}_s = -D_s \frac{\partial \eta}{\partial x} \tag{10}$$

where $\eta$ is land-surface height, $x$ is horizontal distance, and $D_s$ is an effective transport coefficient. The Furbish et al. (2009) probabilistic theory for transport due to particle scattering and settling formulates $D_s$ as

$$D_s = k r_p R_a \overline{N_a \left(1 - \frac{c}{c_m}\right)^2} \cos^2 \theta \tag{11}$$

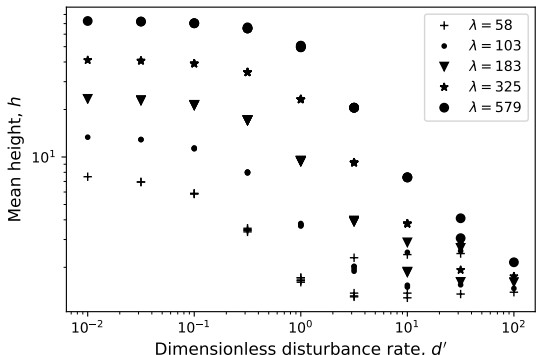

**Figure 9.** Dimensionless mean hillslope height, $h$, as a function of dimensionless disturbance rate $d'$ for a range of hillslope lengths. Data points include 125 sensitivity analysis runs in which $d \in [10^{-3}, 10^{-2.5}, 10^{-2}, 10^{-1.5}, 10^{-1}]$, $\tau \in [10^2, 10^{2.5}, 10^3, 10^{3.5}, 10^4]$, and $\lambda$ as shown in the legend.

where $k$ is a dimensionless coefficient, $r_p$ is particle radius, $R_a$ is active regolith thickness, $N_a$ is the activation rate, $\theta$ is slope angle, $c$ is particle concentration, and $c_m$ is a maximum concentration. The over-bar denotes an average over the active regolith thickness. For the Grain Hill model, $R_a$ scales with the characteristic disturbance depth, $\delta$. Further, because we treat grain aggregates, we may also assume $r_p \sim \delta$. Therefore, we have the prediction that

$$D_s = a\delta^2 N_a \cos^2 \theta \qquad (12)$$

where $a$ is a dimensionless proportionality constant.

The mean expected activation rate, $N_a$, is closely related to the Grain Hill model's disturbance frequency parameter, $d$. To relate the two quantitatively, one needs to make a trivial lattice-geometry correction. A straight-as-possible cut through the hex lattice exposes on average two faces per cell, both of which are susceptible to a disturbance event. Because $d$ is the expected disturbance frequency per cell face, and because independent Poisson events are additive, the resultant disturbance frequency for each cell exposed along a quasi-horizontal surface is $N_a = 2d$.

A more important difference is that whereas $N_a$ is defined as activation rate per unit horizontal area, $d$ represents the rate per unit surface area regardless of orientation. For a given $d$, $N_a$ will increase with surface roughness (because there is more exposed area of regolith-air contact), and with gradient (because the slope length increases).

An additional effect arises from the model's effective $30°$ angle of repose. On slopes steeper than this, the expected disturbance rate increases substantially because gravitational dislodgement becomes activated (Figure 5, bottom row). Thus, the Grain Hill model incorporates an additional nonlinear relationship between flux and gradient inasmuch as $N_a$ depends on gradient.

We can derive an effective diffusivity, $D_e$, from the modeled topography by applying the expected relationship between mean elevation and diffusivity. Here $D_e$ is defined as that value which, if it were spatially uniform, would yield the same mean

steady-state elevation as that produced by the particle model. Framing it this way allows us to interrogate how the effective transport coefficient varies as a function of mean slope gradient. At steady state, mass balance implies that

$$\mathbf{q}_s = Ex \tag{13}$$

where $E$ is the rate of erosion—equal to the rate of material uplift relative to baselevel—and $x$ is horizontal distance from the ridge top. Substituting equation (10) and rearranging,

$$\frac{d\eta}{dx} = -\frac{E}{D_s(x)}x \approx -\frac{E}{D_e}x \tag{14}$$

Integrating and then averaging over $x$, we can solve for the average elevation, $\overline{\eta}$:

$$\overline{\eta} = \frac{E}{3D_e}L_h^2 \tag{15}$$

where $L_h = L/2$ is the length of the slope from ridge top to base (in other words, half the total length of the domain). We can then rearrange this to find $D_e$:

$$D_e = \frac{E}{3\overline{\eta}}L_h^2 \tag{16}$$

To examine how $D_e$ scales, we can define a dimensionless form, normalizing by the disturbance frequency, $d$, and the square of active regolith thickness (equal to particle diameter), $\delta^2$:

$$D_e' = \frac{D_e}{d\delta^2} = \frac{EL_h^2}{3\overline{\eta}d\delta^2} \tag{17}$$

Noting that $E = \delta/\tau$, $L = 2L_h$, and $L/\delta = \lambda$, this is equivalent to

$$D_e' = \frac{\lambda^2}{12\overline{h}d\tau} \tag{18}$$

where $\overline{h}$ is the mean hillslope height in particle diameters.

As expected, $D_e'$ increases with hillslope gradient (Figure 10). The effective diffusivity approaches an asymptote at $30°$ (mean gradient $\approx 0.6$), representing an angle of repose. The pattern resembles the family of nonlinear flux-gradient curves introduced by Andrews and Bucknam (1987) and explored further by Howard (1994) and Roering et al. (1999). At low gradients, $D_e'$ approaches a value of about 60. (This method of estimating $D_e'$ is similar to fitting the standard theoretical parabolic curve to the experimental profiles, except that here we use the integral of the profiles.)

The link between $D_e$ and $d$ provides a way to scale the Grain Hill model to field-derived estimates of $D_s$ and $R_a$. Here we equate the theoretical effective diffusivity, $D_e$, with the definition of the transport coefficient $D_s$ of Furbish et al. (2009). Noting that at low gradients, $\cos^2\theta$ in equation (12) approaches unity, and using the prior relation $N_a = 2d$, we may write $D_s$ for low slope angle as

$$D_s(\theta \to 0) = 2a\delta^2 d. \tag{19}$$

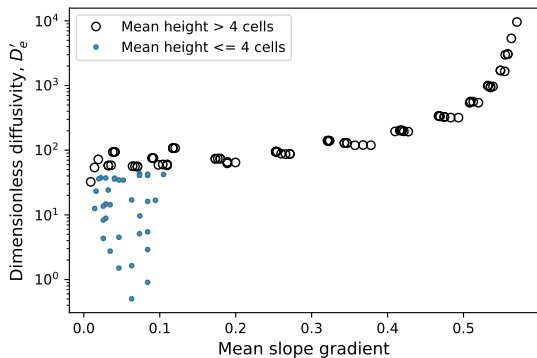

**Figure 10.** Relationship between dimensionless diffusivity and mean gradient, from the series of 125 model runs of which a subset are shown in Figure 8.

In the Grain Hill model, the fact that low-angle $D'_e \approx 60$ implies that the dimensional equivalent $D_e(\theta \to 0) \approx 60\delta^2 d$. Equating $D_s$ (the transport coefficient derived by Furbish et al. (2009)) and $D_e$ (the effective transport coefficient derived from the Grain Hill model),

$$D_s(\theta \to 0) \approx 60\delta^2 d. \tag{20}$$

This relation can be used to scale the parameters in the Grain Hill model with field data. For example, if one were to assume an active regolith thickness of 0.4 m and a low-gradient transport coefficient of $D_s = 0.01$ m$^2$/yr, and set $\delta$ to the active regolith thickness, then

$$d = \frac{D_s}{60\delta^2} \approx 0.001 y^{-1}. \tag{21}$$

Here, $d$ represents the frequency with which a given exposed patch of regolith of width and depth $\delta$ is disturbed upward. With the above values, the simulated hills in Figure 8 would be 232 m long (valley-to-valley) with height ranging from 1.6 to 57.6 m and baselevel lowering rate from 0.04–4 mm/yr.

## 3.2 Hillslope with regolith production from rock

Having established that the Grain Hill model reproduces classic soil-mantled hillslope forms and has parameters that can be related to the parameters in commonly used continuum hillslope transport theories, we turn now to the case in which regolith is generated from bedrock with a production rate that may (or may not) limit the rate of erosion. We explore the role of regolith production with a series of model runs in which $w'$ varies from 0.4 to 40. The upper end of this range represents a condition in which the potential maximum rate of regolith production greatly exceeds the rate of baselevel lowering. The lower end, 0.4, is less than the rate of baselevel fall, and would seem to be insufficient to allow for equilibrium to occur, and yet nonetheless it does. Examples of equilibrium hillslope forms found in this parameter space are shown in Figure 11.

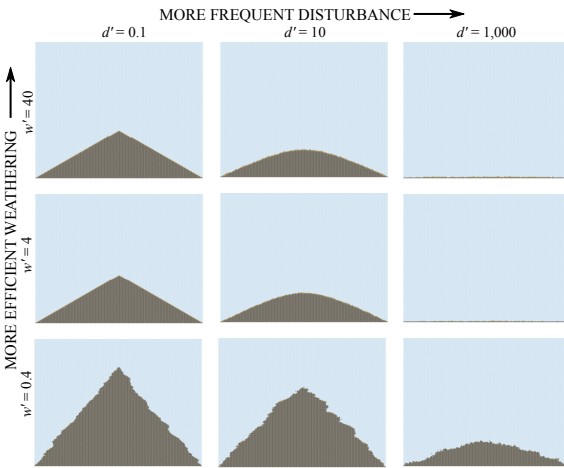

**Figure 11.** Final equilibrium profiles from Grain Hill runs with rock and weathering. Domain size is 222 rows by 257 columns, and uplift interval ranges from 100 to 10,000 years.

Relationships among mean gradient, fractional regolith cover, dimensionless disturbance rate $d'$, and dimensionless weathering rate $w'$ are illustrated in Figure 12. For $w' > 1$, the gradient-$d'$ relation (Figure 12a) has the same shape as in the purely regolith models: a threshold regime at lower $d'$ transitioning to an inverse gradient-$d'$ relation at higher $d'$. This indicates that when the maximum weathering rate (for a flat surface) is substantially greater than the rate of baselevel fall, we recapture transport-limited conditions. With $w' < 1$, however, the hillslope achieves an equilibrium gradient that is greater than that for the transport-limited case, and at lower $d'$, is greater than the threshold angle (Figures 12a,b).

We can also examine the fractional regolith cover, which is defined here as the number of rock-air cell pairs divided by the total number of cell pairs at which air meets either regolith or rock (Figures 12c,d). The fractional regolith cover shows relatively little sensitivity to $d'$ (Figure 12c). The cover hovers around unity for high $w'$ and $d'$, but systematically declines with $w'$ when $w'$ is below about 10. (Note that the data points representing $d' = 1000$ and $w' > 1$ have hillslope heights of only a few particles, and are therefore sensitive to finite-size effects).

The models with $w' < 1$ present a seeming paradox: how is it possible to achieve an equilibrium form when the maximum weathering rate appears to be lower than the rate of uplift relative to baselevel? The solution to the paradox lies in surface area. The surface area of rock that is exposed to weathering is not fixed, but rather depends on the overall slope length, the terrain roughness, and the fractional regolith cover. To appreciate the first effect, consider a planar slope at angle $\theta$ with no regolith cover. If $w\delta$ represents the maximum slope-normal bedrock weathering rate, then the vertical rate is simply $w\delta/\cos\theta$. All else equal, increasing gradient will increase vertical weathering rate, thereby providing a feedback between gradient and rock lowering rate. A second feedback relates to topographic roughness: all else equal, a rougher surface will experience a greater weathering rate because it provides more surface area. The third feedback, which is embedded in the depth-dependent regolith production hypothesis (Ahnert, 1967) lies in regolith cover: the greater the exposure of rock (or the thinner the cover), the

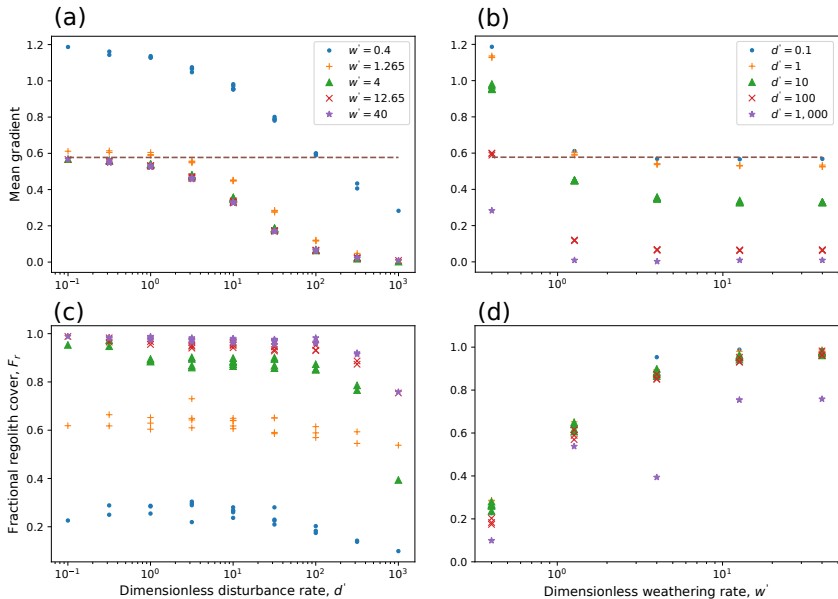

**Figure 12.** Mean equilibrium gradient and regolith thickness for models with rock and weathering, as a function of $d'$ and $w'$. Data represent 125 runs with $d \in [10^{-3}, 10^{-2.5}, 10^{-2}, 10^{-1.5}, 10^{-1}]$ y$^{-1}$, $w' \in [10^0, 10^{0.5}, 10^1, 10^{1.5}, 10^2]$, and $\tau \in [10^2, 10^{2.5}, 10^3, 10^{3.5}, 10^4]$ y.

faster the average rate of rock-to-regolith conversion. In the Grain Hill model, this third feedback is represented by fractional bedrock exposure (since weathering only occurs when rock cells are juxtaposed with air cells).

To test whether these are indeed the feedbacks responsible for equilibrium topography in the Grain Hill model, we can compare the rate of material influx (uplift relative to baselevel) with the expected rate of rock-to-regolith conversion. In the

5  Grain Hill model, the expected rate of regolith production, $P$, in cross-sectional area per time, is the product of weathering rate per cell face, $w$, the cross-sectional area of a cell, $A$, and the number of rock-air cell faces, $n_{ra}$,

$$P = wAn_{ra}, \tag{22}$$

The rate of material addition due to uplift relative to baselevel, $U$, again in cross-sectional area per time, is the area of a cell, $A$, times the horizontal width of the domain in cells, $n_H$, divided by the interval between uplift events, $\tau$:

10  $$U = n_H A/\tau. \tag{23}$$

Equality between rock uplift and weathering can be expressed as:

$$\frac{1}{\tau} = w\frac{n_{ra}}{n_H}. \tag{24}$$

The ratio on the right side represents the surface-area effect, in the form of surface area exposed to weathering per unit horizontal area. The balance is illustrated in Figure 13, which compares the left-hand and right-hand terms for each of the 125

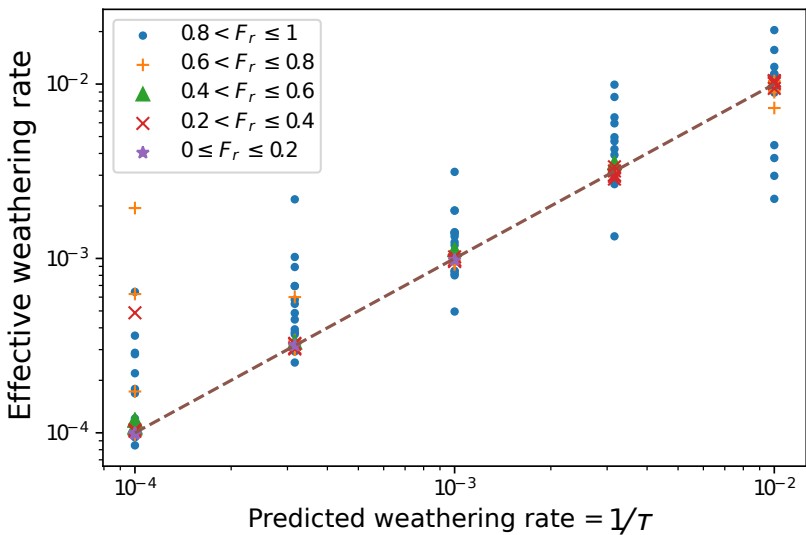

**Figure 13.** Comparison between rate of material input, $1/\tau$ (cells/year), with effective rate of weathering, $wn_{ra}/n_H$, from 125 model runs (see text).

model runs with weathering. Each data point represents a single snapshot in time, and so scatter is to be expected. To help diagnose the scatter around the 1:1 line, the data are divided into quintiles by fractional regolith cover, $F_r$ (note that some of the points in the lower quintiles are obscured by being over-plotted along the 1:1 line). Many of the points that fall off the 1:1 line, especially at the high end (higher $1/\tau$), come from runs with $F_r > 80\%$; with very few exposed rock-air pairs, a small

fluctuation in the $n_{ra}$ can produce a relatively large change in predicted weathering rate. At the low end, many of the points above the 1:1 line come from runs with a maximum height of only a few cells, which are subject to finite-size effects.

The main message of Figure 13 is that the Grain Hill model demonstrates an equilibrium adjustment between rock uplift and rock weathering. The weathering rate does not have a fixed upper "speed limit," but rather is set by the exposed surface area, which in turn is a function of gradient, roughness, and regolith cover. Solutions with a discontinuous regolith cover are

indicative of this adjustment. Slopes can grow arbitrarily steep, with weathering and erosion increasingly attacking from the sides as the gradient rises.

### 3.3    Comparison between weathering rule and inverse-exponential model

The most popular function to describe regolith production from bedrock is the decaying exponential formula proposed by Ahnert (1967), which has proved consistent with estimates of production rate obtained using cosmogenic radionuclides (Heimsath

et al., 1997; Small et al., 1999). The production rate is given by:

$$P = P_0 \exp\left(-R/R_*\right), \tag{25}$$

where $P_0$ is the maximum (bare bedrock) production rate, $R$ is regolith thickness, and $R_*$ is a depth-decay scale on the order of decimeters. On a flat surface, assuming no erosion or deposition, the expected rate of change of $R$ over time is:

$$\frac{dR}{dt} = \frac{\rho_r}{\rho_s}(1-\omega)P_0\exp\left(-R/R_*\right), \tag{26}$$

where $\rho_r$ and $\rho_s$ are the bulk densities of parent material and regolith, respectively, and $\omega$ is the fraction of parent material removed in solution upon weathering. Starting from a bare surface, and assuming isovolumetric weathering (in which case $\rho_s = (1-\omega)\rho_r$), the expected regolith thickness as a function of time can be found by integrating equation (26):

$$\frac{R}{R_*} = \ln\left[\frac{P_0}{R_*}t + 1\right] \tag{27}$$

We can compare this with the behavior of the cellular weathering rule by running the case of a flat, initially bare-rock surface from which weathered material may neither enter nor leave (Figure 14, case $d/w = 0$). When the disturbance rate is zero, the cellular weathering model asymptotically approaches a steady regolith thickness of exactly one cell (thickness = $\delta$). This is so because the model allows weathering to occur only when rock cells are exposed to air cells, and there is no disturbance process that would juxtapose rock and air once the initial weathered layer has formed. When disturbance rate is nonzero, however, regolith continues to form even after the mean thickness $r$ exceeds unity (representing one characteristic disturbance depth). Continuation of regolith production occurs because the disturbance process intermittently exposes rock, at which point it becomes subject to weathering. The greater the disturbance rate, the more frequent the exposure and hence the more rapid weathering (Figure 14). For any ratio $d/w$, the model's weathering behavior clearly differs from the logarithmic growth in thickness predicted by exponential theory. This represents both a strength and a weakness in the Grain Hill model. On the one hand, the model under its present configuration cannot account for rock-to-regolith conversion resulting from processes that penetrate more than one characteristic disturbance depth $\delta$ into the subsurface. For example, the model neglects the possibility that some plant roots may penetrate deeply and contribute to disaggregation, or that an unusually deep freezing front in a cold winter might cause rock fracture and displacement of the resulting fragments (e.g., Anderson et al., 2012). On the other hand, the model honors the likelihood that soil disturbance and regolith production are closely linked processes, rather than being independent: all else equal a greater disturbance rate will tend to produce faster rates of both regolith production and downslope soil movement.

## 3.4  Rock collapse and vertical cliffs

Some rock slopes display a cliff-and-rampart morphology in which a vertical or near-vertical rock face stands above an inclined, often sediment-mantled buttress (Figures 1 and 15). Although common in sedimentary rocks where a resistant unit forms the cliff and a weaker unit the buttress, the same morphology is sometimes found in apparently homogeneous lithology (Figure 15b). The cliff portion of such slopes suggests a process of undermining and collapse, with the cliff-forming material being cohesive enough to maintain a vertical face but too weak to support overhangs.

To explore the origins of ramp-and-cliff morphology, we consider a version of the Grain Hill model that adds an extra rule to represent collapse: any rock particle that directly overlies air has the possibility to transition to a falling regolith particle, with

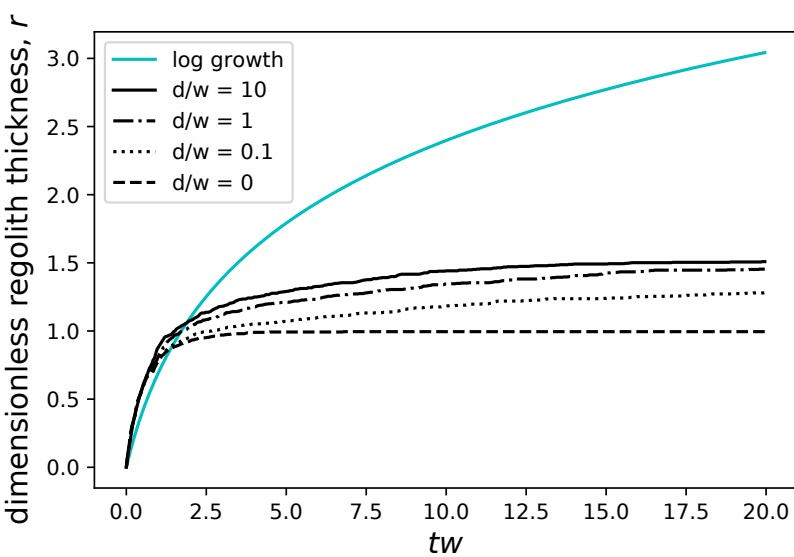

**Figure 14.** Regolith thickness versus time, as predicted by inverse-exponential theory (log growth; solid cyan curve) and the Grain Hill model with a range of ratios of disturbance rate ($d$) to weathering rate ($w$). Time (horizontal axis) is nondimensionalized by multiplying by $w$.

the same rate as gravitational transition from resting to falling—in other words, as soon as a rock particle has been undermined, it behaves like cohesionless material.

Under dynamic equilibrium, this rule produces a morphology with slopes that are roughly planar, with alternating vertical and sloping sections and patchy regolith cover (Figure 16). With $w' \leq 1$, gradient and regolith cover depend strongly on $w'$
and show little or no sensitivity to $d'$. When $w' \ll 1$, the hillslope forms resemble pinnacles. These examples demonstrate two combined feedbacks between weathering and baselevel fall: the surface area susceptible to weathering, and the frequency and magnitude of material collapse through undermining.

The case of transient evolution under a stable baselevel leads to the formation of a regolith-mantled, angle-of-repose ramp (Figure 17). The slope break remains relatively sharp as it retreats headward. The ramp forms as a transport slope. The angle of
repose is an attractor state: if the angle were steeper, weathered material would be rapidly removed as a result of gravitational instability; if it were substantially lower, material would accumulate, because transport would be limited to the (much lower) rate of disturbance-driven creep motions. Hence, the Grain Hill model predicts that formation of a sediment-mantled ramp beneath a steeper, actively weathering rock slope is an expected outcome for a steep rock slope under stable baselevel.

### 3.5 Blocks

Weathering and erosion in landscapes underlain by relatively massive, fracture- or joint-bounded rock can sometimes produce large "blocks" of rock, defined here as clasts that are too large to be displaced upward by normal hillslope processes. The release

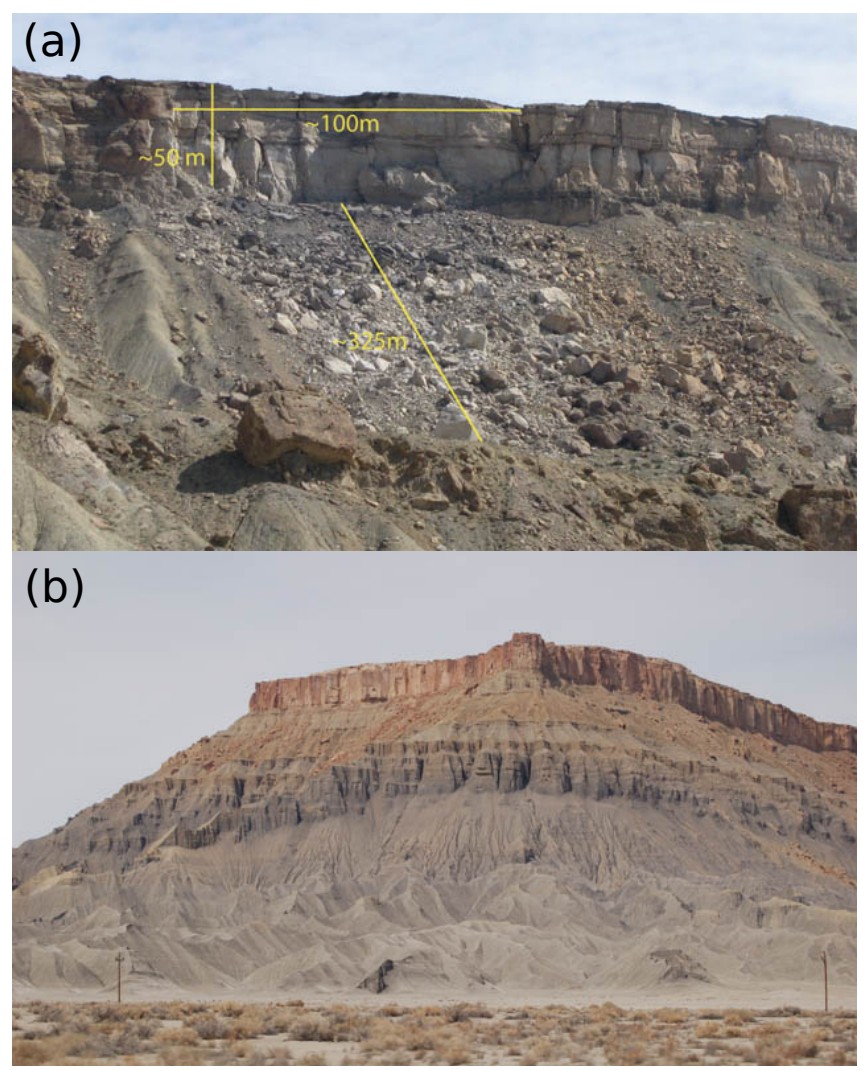

**Figure 15.** Two examples of cliff-and-rampart morphology. (a) Near Palisade, Colorado, USA, after recent rock-fall event (photo courtesy D.N. Bradley and D. Ward). (b) Colorado Plateau, Utah, USA. Note that contact between lower rampart and sub-vertical slopes, both of which have formed in a gray shale unit, occurs without any apparent break in lithology.

of blocks from dipping sedimentary or volcanic strata can alter both the shape and relief of hillslopes (Glade et al., 2017). When blocks are delivered to streams, they can influence the channel's roughness, gradient, erosion rate, and longitudinal profile shape (Shobe et al., 2016).

As discussed in Section 2.6, the Grain Hill model can be modified to honor blocks by defining an additional cell type that represents blocks. The weathering process is modified such that a rock cell now weathers into a block, and the block in turn

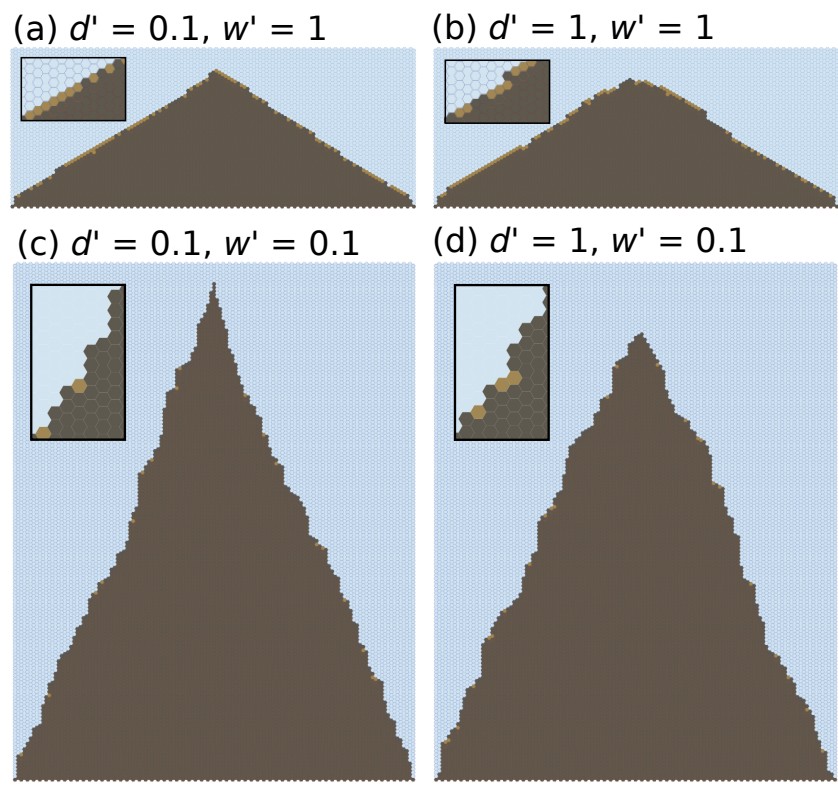

**Figure 16.** Quasi-steady model hillslope profiles created using a collapse rule, under four different combinations of $d'$ and $w'$. Insets show magnified views of a portion of each hillslope.

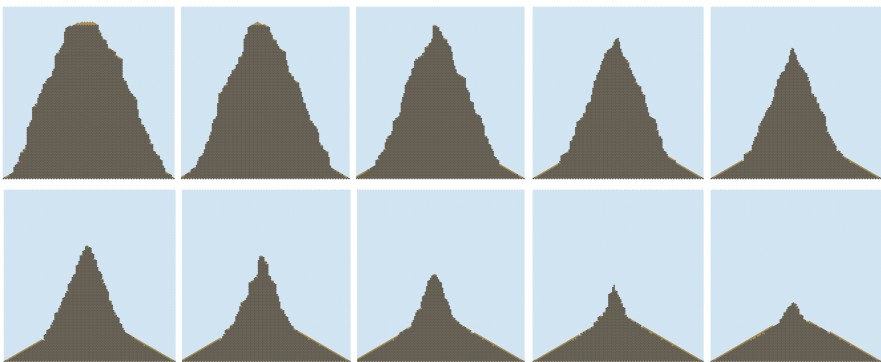

**Figure 17.** Time series showing transient erosion of a steep rock slope under a stable baselevel, highlighting formation of ramp-and-cliff morphology. Simulation shows 20,000 years of slope evolution under $d = w = 10^{-3}$ y$^{-1}$. Nominal width, assuming $\delta = 0.1$ m, is 12 m.

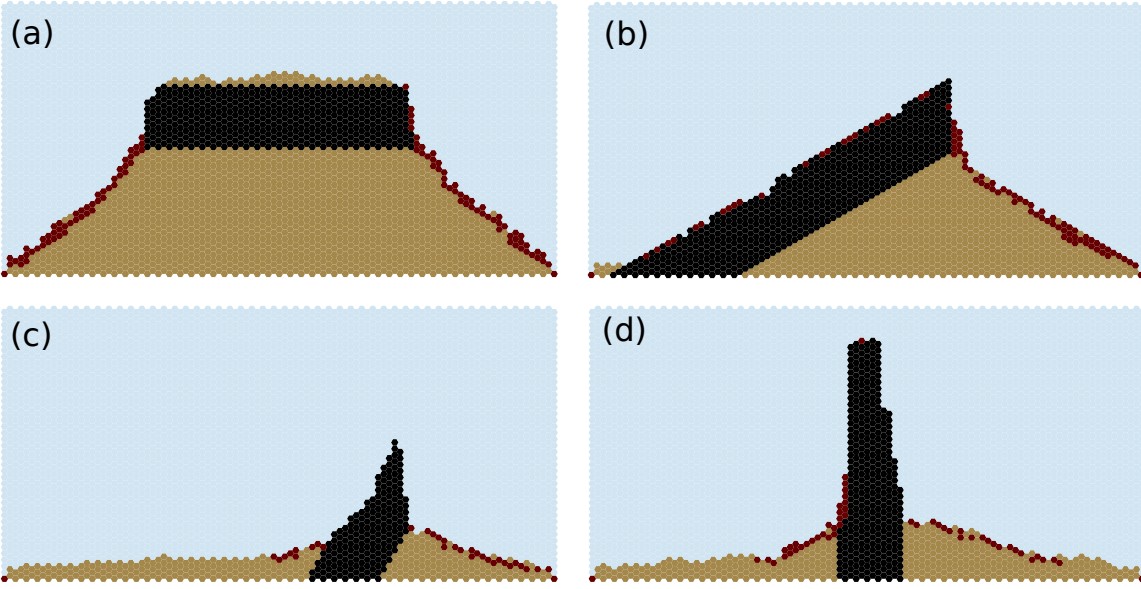

**Figure 18.** Examples of models that include blocks. Rock (black) weathers to blocks (dark red), which can only move downward or downward-plus-laterally. Blocks in turn weather to regolith (light brown).

may weather to form regolith. When a block is undermined directly from below, it will fall just as a normal regolith particle would. When a block particle lies adjacent to and above an air cell, a disturbance event may occur that causes the block to shift downward on the slope. By these means, blocks in the model may move downward or downward-and-laterally, but never upward. An implicit assumption in this treatment is that blocks do not roll long distances (further than their own diameter) upon release.

We examine model runs in which a resistant rock layer is embedded in a weak sedimentary material that is soft enough to be treated as regolith (Figure 18). The modeled hillslopes are qualitatively consistent both with field observations and with the mixed continuum-discrete model of Glade et al. (2017) and Glade and Anderson (2017) in that block-mantled slopes are generally concave-upward, reflecting a downslope decrease in the flux of blocks as weathering progressively transmutes them into regolith.

## 4 Comparison to field sites

We perform a basic validation of the Grain Hill model by comparing its output to real field sites, testing whether the model is capable of reproducing realistic hillslope forms at the correct spatial scale under known boundary conditions. Field sites were chosen such that model boundary conditions could be derived from independent field estimates of rate parameters such as $D_s$ and the rate of baselevel fall. To perform this test, we consider two examples: a convex-upward, soil-mantled hillslope

in Gabilan Mesa, California, USA (Figure 2a,b), and a steep, quasi-planar, discontinuously mantled hillslope in the Yucaipa Ridge, California, USA (Figure 2c,d). For each of these two case studies, the hillslopes appear to be approximately at steady state, and independent estimates exist for the rate of baselevel fall, $U$ (Binnie et al., 2007; Perron et al., 2009, 2012). We estimated the effective transport coefficient, $D_s$, for the profiles shown in Figure 2a,c by measuring the second derivative of the one-dimensional hillslope elevation profiles, $\frac{\partial^2 \eta}{\partial x^2}$, and solving for $D_s$ using

$$D_s = -\frac{U}{\frac{\partial^2 \eta}{\partial x^2}}. \tag{28}$$

For the Gabilan Mesa profile, we estimated the profile-averaged effective transport coefficient as 0.0345 m$^2$y$^{-1}$. The effective rate of baselevel lowering has been estimated at $U \approx 1.47 \times 10^{-4}$ m y$^{-1}$ (Perron et al., 2012). To construct a Grain Hill model for the Gabilan profile, we begin by assuming a characteristic disturbance depth of $\delta = 1$ m. This value was chosen to be consistent with measured soil depths that typically range between 0.2 and 1.2 m (Johnstone et al., 2017). We treat the system as transport-limited, consisting of mobile material, so that weathering is not explicitly modeled. The disturbance parameter, $d$, is then calculated from the independently estimated value of $D_s$ using equation (20). The interval between uplift events is $\tau = \delta/U \approx 6800$ y. The resulting modeled equilibrium profile provides a reasonably good match to the observed Gabilan profile, with a convex-upward shape and a hilltop height of about 45 m above the slope base (Figure 19a).

For Yucaipa Ridge, we estimated the transport coefficient at $D_s \sim 0.028$ m$^2$y$^{-1}$ on the basis of hilltop curvature and an estimated effective rate of baselevel lowering of $\approx 0.0027$ my$^{-1}$ (Binnie et al., 2007). Using equation (20), this equates to a disturbance-rate parameter $d = 0.00468$ y$^{-1}$ and an uplift interval of 370 y in the Grain Hill model. Bedrock outcrops are common on the Yucaipa Ridge hillslopes, implying a thin, discontinuous regolith cover. We therefore treat the system as consisting of bedrock that must be weathered before it can become mobile. Because we do not have independent information on the effective maximum rock weathering rate, the Yucaipa case is a somewhat weaker test: we can only ask whether there exists a geologically reasonable value of $w$ such that the model reproduces the observed relief and shape of the slope profile. Through trial and error, we find that with a weathering rate parameter $w = 0.002$ y$^{-1}$ (which corresponds to a maximum regolith production rate of 2 mm y$^{-1}$), the model does a credible job of capturing the shape and size of the Yucaipa profile (compare Figure 2c with Figure 19b). Although this particular value was obtained through a simple calibration process, it is at least both geologically reasonable and, as one might expect, somewhat lower than the rate of baselevel lowering.

To test the sensitivity of the Yucaipa example to the assumed characteristic disturbance depth, we ran a second experiment in which $\delta$ was reduced to 0.75 m, and the weathering, disturbance rate, and uplift parameters were re-scaled to accordingly. The relief and mean gradient of the two cases are nearly identical, with planar slopes and a relief in both cases of $\sim 100$ m.

These two examples demonstrate that the Grain Hill model parameters are not arbitrary, but instead can be linked through straightforward reasoning to field estimates of transport efficiency and baselevel lowering. When one does so, the model successfully reproduces both the shape and scale of observed slopes.

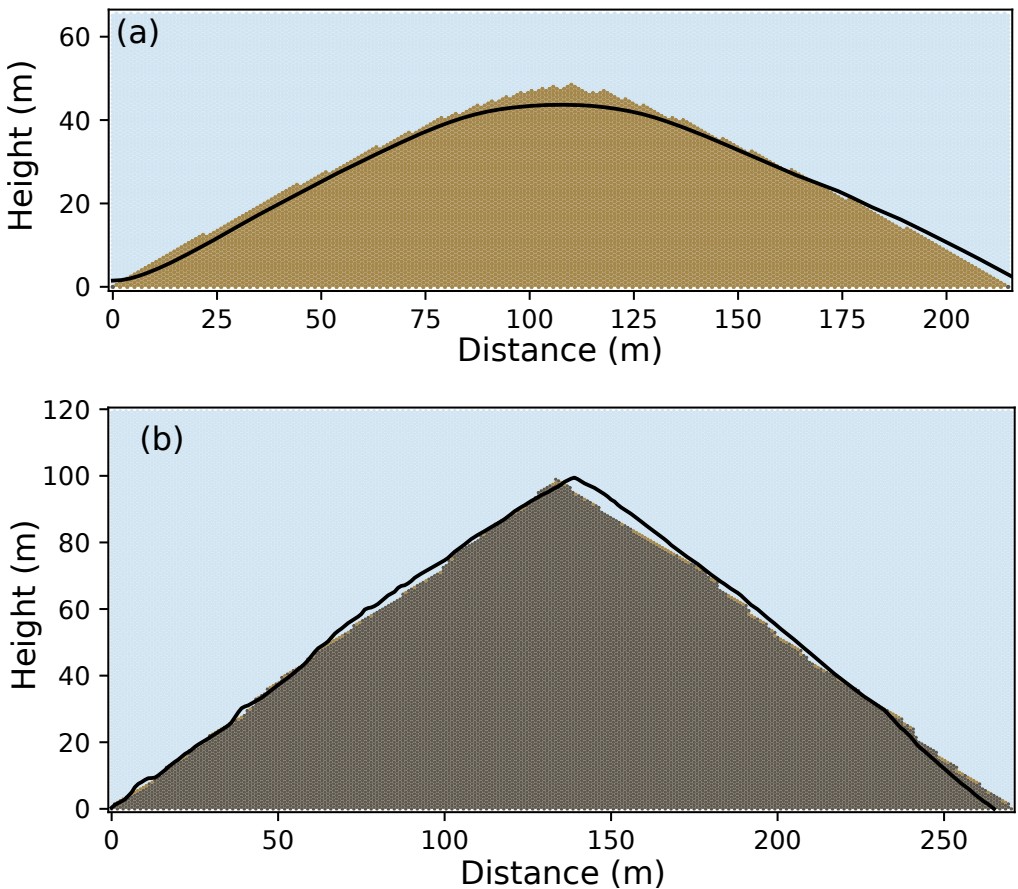

**Figure 19.** Steady state models using parameters estimated from observed hillslope profiles. (a) Parameters based on Gabilan Mesa, with the profile shown in Figure 2a,b for comparison. (b) Parameters based on Yucaipa Ridge, with the profile shown in Figure 2c,d for comparison.

## 5   Discussion

With just three parameters—disturbance frequency ($d$), characteristic disturbance depth ($\delta$), and baselevel fall frequency ($u$)—the Grain Hill algorithm can reproduce the convex-upward to quasi-planar forms associated with soil-mantled hillslopes (Figure 8). With the addition of a parameter that represents rock-to-regolith conversion rate, the algorithm accommodates partly

5   mantled, rocky hillslopes (Figures 11, 16, 17). By adding a rule for detachment of blocks from resistant rock, the model reproduces hillslope forms associated with hogbacks and ledge-forming escarpments (Figure 18).

A common criticism of cellular automaton models is that they involve arbitrary rules and/or parameters that can neither be measured nor verified in the real world. That is not the case for the Grain Hill model, for which the parameters are tied to measurable physical quantities. For example, the disturbance frequency $d$ is directly related to the frequency parameter $N_a$ in

10   statistical theory of soil transport developed by Furbish et al. (2009), and through that theory to the diffusion-like transport

coefficient $D_s$ that is commonly estimated in field studies. This connection between model parameters and field measurements is illustrated by the model's ability to reproduce the correct shape and scale of observed hillslope forms when estimates of $D_s$ and $U$ are available (Figures 2, 19). In the transport-limited case, there are no tunable parameters: given independent estimates of $D_s$ and $U$, the correct morphology is recovered (Figure 2a, 19a). In the case where rock weathering appears to play a role, and an independent estimate of $P_0$ is not available, the model requires an estimation of maximum weathering rate $w$. Nonetheless, a plausible value of $w$ (0.002 m y$^{-1}$), somewhat smaller than the rate of baselevel fall (0.0027 m y$^{-1}$), reproduces the observed shape and relief in the Yucaipa Ridge case study.

The transport dynamics predicted by the Grain Hill model are consistent with continuum soil-transport theory, which treats soil as a fluid with a downslope flow rate that depends on slope gradient. Like the popular Andrews-Bucknam nonlinear transport law (e.g., Andrews and Bucknam, 1987; Howard, 1994; Roering et al., 1999), the transport-limited form of the Grain Hill model predicts diffusion-like behavior in which the effective diffusivity increases with slope gradient, with an asymptote at a threshold angle (Figure 10). In one sense, the Grain Hill model is actually closer to the process level than fluid-like continuum models, because net downslope mass flux arises from a sequence of stochastic disturbance events rather than being dictated by a macroscopic transport law.

One limitation of the Grain Hill model is that its threshold-like behavior arises from the lattice geometry: regolith cells perched at a 30° angle above and to one side of an air cell are treated as unconditionally unstable. Whereas the timing of motion is treated as a stochastic process, the occurrence of motion is inevitable (unless some other event occurs first). This treatment neglects the possibility of frictional locking among noncohesive grains at angles somewhat above 30°, as well as the possibility of cohesion. This limitation could be overcome by introducing a probabilistic treatment of grain stability: a grain aggregate will be stable with a given probability $p$, and unstable with probability $1 - p$. Such a treatment would introduce an additional parameter, but this parameter could in principle be estimated from physical experiments. The addition of a "sticking rule" like this might also make it possible for models with alternative lattice geometries to manifest the same dynamics, thereby de-coupling the basic model framework from the geometry of the lattice on which it is implemented.

The inclusion of rock-to-regolith conversion enables the Grain Hill model to predict a continuum of slope forms from fully soil mantled to intermittently covered to bare. However, there are several limitations in the treatment of regolith production that could be improved on. The weathering rule assumes that regolith production can only occur when rock is exposed to air, which obviously neglects the role of shallow subsurface processes such as root or frost wedging. The effective weathering depth scale is the same as the disturbance scale, and equal to the cell size. This assumption is probably reasonable if the processes responsible for weathering and disturbance were one and the same, but not if they are distinct processes with different length scales. The Grain Hill model also does not account explicitly for chemical weathering, which in some cases can extend well below the surface. Finally, the model's effective regolith-production behavior does not follow the log-growth curve predicted by inverse-exponential theory for a stable surface (Figure 14). With these caveats in mind, one advantage of the stochastic model of regolith production is that it effectively treats the disturbance and regolith-production processes as being closely linked: all else equal, the production rate is higher when disturbance is more frequent.

The popular inverse-exponential model for regolith production implies the existence of a speed limit to landscape evolution: in the absence of rock landsliding, erosion rate cannot exceed the maximum rate of rock-to-regolith conversion. Moreover, the model implies the existence of a bare landscape once the rate of erosion exceeds the maximum rate of regolith production. Heimsath et al. (2012) found evidence, however, that in fact there are additional stabilizing mechanisms, and that these manifest in landscapes with thin, patchy soils. The Grain Hill model is consistent with these observations in that it predicts the natural emergence of a discontinuous regolith cover, with the fractional cover exerting an influence on the average rate of weathering and erosion. Furthermore, the model behavior highlights the importance of slope length and roughness in modulating the regolith production rate: all else equal, steeper or rougher slopes allow higher production rates, leading to an additional feedback between relief and erosion rate for rocky hillslopes. The possibility of rock collapse upon undermining by weathering provides another feedback mechanism that may allow rates of erosion to exceed the flat-surface maximum regolith production rate (Figure 16).

The Grain Hill model also provides insight into transient evolution of rocky slopes. Experiments on the relaxation of rocky slopes that are steeper than the threshold angle predict the formation of a regolith-mantled pediment at the angle of repose, which extends upslope as the steep upper slope gradually recedes (Figure 17). This scarp-pediment morphology emerges without any variation in material strength, requiring only a period of baselevel stability.

As a computational framework for exploring hillslope forms, the Grain Hill model has the advantage that it provides a mechanistic link between events (disturbance and weathering) and long-term morphologic evolution, without the need to specify a flux law. The model has the further advantage of being fully two dimensional, allowing disturbance and weathering events to initiate from the side as well as vertically. A further key element is that the model can mix timescales: a short timescale associated with grain motion, an intermediate time scale associated with disturbance events, and a much longer timescale for slope evolution. Mixing these disparate timescales in a single computer model is made possible by the fact that most of the time grains are stationary: the algorithm operates on small (stochastic) time steps during those moments when grains are moving, and on much longer steps when no grains are in motion (for further information on the discrete-event algorithm behind the model, see Tucker et al. (2016)).

The Grain Hill framework has several important limitations. It is not practical to simulate motion of individual grains unless the spatial scale is quite limited (e.g., Figure 6) or the grains are unusually large (Figure 18). If one wished to model individual grains (of order say $10^{-3}$ m) at the scale of a hillslope (of order $10^2$ m), a much more efficient solution algorithm would be needed. Furthermore, the nature of a cellular automaton is such that physical interactions are limited to adjacent cells only; long-distance effects such as stress transmission cannot easily be represented. In one sense, the restriction to short-range influence could be seen as an advantage, in that it forces one to think about how it is that mass or energy is actually transmitted in a granular medium. But the restriction means that well-known principles such as solid-state stress cannot easily be represented. On the other hand, the model does capture non-local transport, in which particles set in motion can travel a distance comparable to the slope length (Foufoula-Georgiou et al., 2010; Tucker and Bradley, 2010; Furbish and Roering, 2013). Nonlocal transport emerges in the Grain Hill model when the slope angle is near or above $30°$, such that there is a high probability that a disturbed particle will land in an unstable location and continue moving without the need for a second disturbance event.

A further limitation concerns the fixed cell size. Because the model is restricted to a fixed cell size, the Grain Hill framework does not lend itself to treatment of multiple grain sizes (apart from the simple "aggregates and blocks" approach illustrated in Figure 18). Despite these limitations, the Grain Hill model provides a useful framework for exploring hillslope process and form in the context of stochastic events.

## 6   Conclusions

A continuous-time stochastic cellular automaton model known as the Grain Hill model allows for computational simulation of two-dimensional slope forms that arise from stochastic disturbance and (possibly) weathering events. The model operates on a hexagonal lattice, with cell states representing fluid, rock, and grain aggregates that are either stationary or in a state of motion in one of the six cardinal lattice directions. An optional additional state represents unusually large grains ("blocks") that cannot be displaced upward by disturbance events.

The Grain Hill model is able to reproduce a range of common slope forms, from fully soil mantled to rocky and partially mantled. The bestiary of forms that the model can produce includes convex-upward soil mantled slopes, planar slopes (bare, soil mantled, or in between), and cliffs with basal ramparts. When the model is configured to include a resistant rock layer that decomposes into blocks, the model reproduces observed hogback-like slope forms and qualitatively matches the behavior predicted by a recent continuum-discrete model (Glade et al., 2017; Glade and Anderson, 2017).

In its simplest guise, the model has only three process parameters, which represent disturbance frequency, characteristic disturbance depth, and baselevel lowering rate, respectively. Incorporating physical weathering of rock adds one additional parameter, representing the characteristic rock weathering rate. These parameters are not arbitrary but rather have a direct link with corresponding parameters in continuum theory. Comparison between observed and modeled slope forms demonstrates that the model can reproduce both the shape and scale of real hillslope profiles.

Experiments with the Grain Hill model highlight the importance of regolith cover fraction in governing both the downslope mass transport rate and the rate of physical weathering. Equilibrium rocky hillslope profiles are possible even when the rate of baselevel lowering exceeds the nominal bare-rock weathering rate, because increases in both slope gradient and roughness can allow for rock weathering rates that are greater than the flat-surface maximum. Finally, experiments in transient relaxation of steep, rocky slopes predict the formation of a regolith-mantled pediment that migrates headward through time while maintaining a sharp slope break.

*Author contributions.*   The idea to develop a 2D cellular rock-slope model arose from conversations among all three authors. The model code was written in Landlab by GT. Both GT and DEJH contributed to the underlying grid data structures and Python code. SWM extracted the hillslope profiles and estimated the parameters for the two field sites. GT performed the computational experiments and wrote the paper, with input and editing from SWM and DEJH.

*Acknowledgements.* This research was supported by the US National Science Foundation (EAR-1349390 and ACI-1450409 to GT and DEJH, and EAR-1349229 to SWM). DEJH's participation was also supported in part by the National Center for Earth Surface Dynamics (EAR-1246761). Support for high-performance computing and software development was provided by the Community Surface Dynamics Modeling System (CSDMS) (EAR-1226297). High-resolution topographic data was downloaded from Open Topography (http://www.opentopography.org/).

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
