# Peer review of "A lattice grain model of hillslope evolution"

_Earth Surface Dynamics, 2018_

## Referee Comment (RC1) · E. Foufoula-Georgiou (Referee) · 14 Mar 2018

**Review of "A lattice grain model of hillslope evolution" by Tucker, McCoy and Hobley (*ESURF*)**

**Short-summary:**

The authors present a two-dimensional *reduced complexity* model of hillslope evolution called The Grain Hill Model. This model is based on the lattice-grain model introduced by Tucker et al. (2016), which is a continuous-time cellular automaton model with process-based rules governing the model dynamics and with parameters related to measurable physical quantities. The authors show how the Grain Hill Model is able to reproduce a range of common slope forms, from fully soil mantled to rocky and partially mantled. The hillslope morphologies that the model can reproduce include convex-upward soil mantled slopes, planar slopes, cliffs with basal ramparts, and hogback-like slope forms.

Furthermore, by adjusting the parameters of the model to describe two field-site hillslope examples, the authors show that the model is able to reproduce correctly the form and scale of the landforms.

**Strengths:**

- Overall the paper is interesting. A relatively simple model that is able to integrate very different time scales (short time scale associate to grain motion, intermediate time scale associate with disturbance events and a much larger time scale for slope evolution) and to reproduce a wide range of hillslope morphologies.
- The Grain Hill model is implemented in the Landlab modeling framework, which is open source.
- The authors provide good insights on the physical meaning of parameters and main mechanisms of landform evolution.
- Nice exploration of the parameter space showing some classical examples and providing physical insight as well.

**Weaknesses**:

The main criticism on the paper relates to the model description. The authors present The Grain Hill model in Section 2 ('Model description'). This model builds on the previously introduced (slightly modified) CTS Lattice Grain model (Tucker et al., 2016) which is the main dynamical component of Grain Hill determining the movement of the grains on the hillslope. Although the authors have made some efforts to briefly describe this model component, I have found it insufficient. I am not claiming that the authors should provide all the details (again) of this model, since that description was the object of a previous publication, but given that the scope of the paper is introducing a new model for hillslope evolution (e.g. as indicated by the title), this paper should be sufficiently self-contained. What I mean is that the reader should have a good understanding of the functioning of the model (rules and dynamics) without the need of going back and forth between this manuscript and Tucker et al. (2016), which I have found myself doing quite often. Obviously, the technical implementation of the model rules (e.g., smart algorithms to rank times or how to take advantage of the Landlab's grid architecture) can be omitted, since they are well described in the previous publication, and it is not detrimental to conveying the essence of the model.

In fact, even after going back and forth between this manuscript and Tucker et al. (2016), there are a few issues that remain unclear to me and should be addressed in the manuscript. So please see the following comments and questions:

- My understanding of the grain motion for the most basic grain dynamics of the model is as follows: **(1)** Disturbance is described as a Poisson process, assigning a state transition at a given time for every surface grain (random variable taken from an exponential distribution with parameter $d$), which forms a queue sorted by the transition times. **(2)** The "soonest transition" (change in the state of the corresponding grain) in the queue is executed, which may trigger future state transitions of neighboring grains. **(3)** The gravity and inelastic collision rules (Figure 4 and 5) would dictate the subsequent transitions and movements. I assume that the two latter processes (driven by gravity and collision) operate at smaller time scales. Is this so? If yes:
    - What is the time scale of the gravitational transitions illustrated in figure 5? is that time scale dictated by an additional parameter/rate $g$ (not listed in this manuscript)? If yes:
        - is $g$ a constant rate, and therefore all the grains transition exactly at time $1/g$ after being updated to their current state? or is $g$ the rate that characterizes an exponentially distributed random variable $\tau$, and each grain transitions after a (different) lag $\tau$?
    - What is the time scale at which the inelastic collisions and movements illustrated in figure 4 take place? Same than the previous time scale, $1/g$? constant lag or stochastic random variable?

- My understanding is that when a grain (or aggregate) moves to occupy an empty cell (fluid), it keeps the same state of movement (arrow direction). Is that correct? If yes, can this give rise to grains that temporally float in the fluid (grain surrounded by six fluid cells)? (I think it depends on the previous questions about the characteristic time scales of gravitational and movement/collision transitions).
- How is the model initialized? For example, what is the geometry (flat?)? Do the lateral edges of the grid play a main role on the early evolution of the hillslope?
- How does the grid type (squares vs. hexagons) affect the results? It is stated in page 7 (lines 12-13) that the gravitational rule sets effectively the angle of repose to 30 degrees. Does the angle of repose strongly relate to the chosen hexagonal grid?
- I would really appreciate as a reader to have access to some videos (as supplemental information) of the evolution of the model, including the evolution of the hillslope as it approaches its steady state.
- Figure 3 is not very straightforward. Maybe it would be good to highlight the cell-pair affected by the transition/movement. With the current caption and text, it is confusing to see several arrows, but only one cell pair changing (this takes me back to my first questions about time scales). Also, I guess that this picture is not illustrative of the The Grain Hill model (otherwise some grains would be floating on the air/fluid). Maybe it would be more interesting to provide an example relevant to hillslopes.
- The authors illustrated in Figure 19 that the model is capable of reproducing realistic hillslope forms with parameters estimated from field data. However, it is not straightforward to evaluate the model performance in the absence of a direct comparison with the observations. I suggest that the authors can either (a) overlap Figure 19a,b with the profiles shown in Figure 2a,b to show that qualitatively the model is able to reproduce the form and height of those landforms, or (b) provide

a quantification of the fit of the observed profile with the model results. Regarding the determination of the parameters, it may also be helpful if the authors can comment on their assumption of $\delta$=1 m, as they suggested that the soil depths typically range between 0.2 and 1.2 m which is almost an order of magnitude difference for the variation. Are the results sensitive to this parameter?

Minor points:

- Figure 7 looks disproportionally big in comparison with other figures (e.g. Figure 4 or 5).

- Please label the different panels of Figure 8 (e.g. a,b,…) to make easier their reference in the text.

- The authors state that Figure 9 shows the results of 125 model runs. It is not clear what those different runs are. From the figure, I can see 5x9 different combinations of $\lambda$ and $d$ so, do you run some cases twice and some case three times?

- Figure 19 caption: "State state models" --> "Steady state models".

- Given the scope of the paper and the number of figures, I am not sure if Figure 6 adds much to the explanation of the model.

- Please be consistent in terminology (e.g. mean gradient vs. slope gradient)

Other comments:

- The authors emphasize the simplicity of the model by pointing out the reduced number of parameters (3 in its simplest version). However, this kind of assessment is a little tricky when we refer to CA models, where the type and number of rules play a significant role in evaluating the simplicity of the model.
- For me it is a little contradictory using "continuous-time" and "Cellular Automaton" as descriptors of a model. Cellular automata were originally defined as discrete models both in time and space, as well as the state variable of each grid cell. Models with discrete space and state variables, but continuous time, sometimes are called Boolean Delay Equations (Figure 2 in Zaliapin et al. 2003).

Alejandro Tejedor, Zi Wu and Efi Foufoula-Georgiou

[Figure]

Reference:

I. Zaliapin, V. Keilis-Borok, and M. Ghil, A Boolean delay equation model of colliding cascades. Part I: Multiple seismic regimes, J. Stat. Phys. 111:815-837 (2003).

---

## Referee Comment (RC2) · J. Roering (Referee) · 9 Apr 2018

This manuscript describes a new method to simulate hillslopes based on a lattice-based cellular automaton framework. This work is highly innovative, clever, and wonderfully presented and I recommend publication given some minor considerations and clarifications that I'll detail below.

First, though, I'll note that this manuscript is situated on the cusp of invigorated efforts to revisit how our community does the accounting of mass on hillslopes. The continuum perspective that pervades so much of geophysical theory and simulation is being revisited to account for stochastic, nonlocal, and other effects and the approach adopted here complements forward-looking work by Furbish, Ganti, and others. While cellular automaton schemes have been around for some time, I am not aware of studies that are so carefully crafted as this one as to allow for comparison and calibration

[Figure]

with commonly used continuum parameters. This manuscript offers a clear roadmap for applying CA schemes to well-trod field sites and characteristic landforms.

Second, this manuscript is innovative not only for the algorithmic advances but the interpretations that emerge are also highly compelling. In particular, the ability of the model to account for disturbance processes as the source of both soil transport and production is conceptually appealing and a long-overdue advance. The continuum requirement of breaking up these processes has always been unsatisfying and this effort is to my knowledge the first to provide a unifying treatment. Also, the model's ability to account for retreating cliff-rampart and rocky landforms is highly compelling. The authors neatly applied their toolkit to two field sites as a means to independently confirm their scaling arguments for the model parameters.

Third, (I'm still in gushing mode), I'll note that the lattice grain model presented here is built on parsimony from the parameterization standpoint, which enables the authors to efficiently study a wide range of model outcomes.

Below, I detail several comments for the authors to consider.

1) While the authors claim to invoke minimal parameters in defining their simulations, it seems that the choice of hexagon elements (which conveniently pack purely) as well as the myriad rules for different geometries constitute choices that impact the model outputs. For example, the hexagon packing results in a 30 degree angle of repose slope due to the geometry of the hexagon contacts as well as the gravity rules. This is an arbitrary choice that's convenient for geometry and computational purposes. It turns out that 30 degrees is a fairly common friction angle for smooth cohesion grain piles, so this is appealing. That said, what prevents the model from being performed with octagons? In that case, the packing isn't as efficient and voids are required but it seems defensible nonetheless. Such a geometry would result in a 45 deg friction angle if the same rules were applied. Other element shapes and packing schemes are possible, in fact, a mind-boggling array could be tested. How does the choice

of hexagons affect the results? Beyond setting the angle of repose, how do different packing schemes affect the rules for state changes (see below)?

2) The rules of state changes are nicely laid out with an example orientation in figure 4, although it seems that some possible states are not represented and it would be nice if the figure was comprehensive given that no standards are offered as to why certain motion orientations result in motion and others do not. Can the authors write criteria for the outcomes of the inelastic collisions shown in Figure 4? In particular, the two particles that come together (far right) generate split 50/50 outcomes, why doesn't the same type of outcome emerge with the far left shear-like interaction? I suspect that some of these details don't matter terribly much but it'd be interesting to better understand which do matter and thus account for the behavior observed in the simulations. Is it possible to do an accounting of the state transformations and where they occur in order to determine which interactions dominate in the physical space of the model? Such an exercise is probably beyond the scope of this paper but it'd be very interesting to better understand the impact of the interaction choices listed here.

3) I'm unclear on how the state transition occurs during a given time step. Given that state rules are only defined pair-wise, how are the transitions generated given that each element has 6 neighbors? How is precedence or primacy established? I'm certainly missing some aspect of how these calculations are performed, so perhaps a bit more explanation on this point is possible? Do the pair-state calculations actually occur in sequence? Or does the state transition happen simultaneously? If so, how are the particular pairs chosen? Or is every combination assessed?

4) While the text addresses previous some work on soil-mantled slopes and transport models, the description of rocky slopes and cliff-rampart settings is lacking in scholarship and context. There's a rich literature about cliff retreat in which conceptual models posed are likely consistent with the results generated here. Many workers, including Davis, King, Oberlander, Twidale, Ollier, and others have contemplated these landforms with varying process models in mind, including and notably overland flow. Only

a very recent paper is cited here. The fact that the lattice grain model can generate iconic landforms without directly invoking overland flow is compelling and worth mention and/or context.

5) The emergence of steady state rocky landforms for baselevel lowering rates that exceed the maximum soil production rate is perhaps the most attention-grabbing result of this model. The authors nicely describe how this results from slope and roughness, which is intuitive upon reflection. Given the importance of surface area for the total weathering flux, do any characteristic wavelengths emerge along the interface? is the roughness isolated to the element-scale or is it superimposed on broader wavelength fluctuations? It would be interesting to see if complexity (1/f noise or otherwise) of the surface emerges due to the need for weathering rates to keep pace. The possibility for feedbacks seems like fertile territory.

6) The connection between disturbance and weathering through exposure is very appealing and nicely stated. In this sense, the creation of porosity and the evolution of that porosity in the regolith is a major conceptual advance. That said, it isn't not clear to me in the text that porosity can be essentially advected to depth given a particular sequence of phase state changes. Is that the case? I'm not sure how else porosity goes downward, but however it does, it's worth clarification b/c this sets the weathering rate along the bedrock interface. This goes back to the choices about pair-wise interactions. Would other choices for the interaction rules result in more/less porosity advection?

minor comments: 1) abstract: line 6: the concept of disturbances is part/parcel of this contribution and may merit mention in the abstract. Otherwise, the large block comment is unclear and is probably secondary in importance. 2) pg 2, line 38ish: is it worth mentioning that the lattices don't actually move but rather than the state evolves? it took me awhile to come around to this realization and it could be stated earlier in my opinion. 3) pg 4, line 20-25: the concept of state is employed here before it's explained, I recommend moving some of the text from below (30-33 or so) upwards

to clarify. 4) pg 7, line 22: given the porosity, would it be possible to include water and a weathering function in the future? 5) pg 10, line 16: given comments above about hexagon and friction angle, isn't there really more than four parameters? I'm not sure how the counting works, but my sense is that the hillslope relief should depend on the angle of repose which results from hexagon worldview. 6) pg 13, figure 8: would it be possible to fit the standard paraboblic (and or nonlinear) analytical curves to these experimental profiles? that would be another way to determine the parameter scaling. 7) pg 14, eqn 13: I'm dim, but why is 3 on the denominator rather than 2? My apologies but whenever I've done this integration I end up with 2 on the bottom...please help. 8) pg 19, figure 13: it's difficult to see the various symbols here. 9) pg 20, line 4: isn't the creation of porosity and it's downward advection akin to a depth-dependent process? the text here seems restrictive. 10) 7) The diffusivity values for Gabilan Mesa seem quite high and I wonder if this results from K calibration using low-curvature hilltops that are not reflective of integrated erosion in that site.

---

## Author Comment (AC1) · 21 May 2018

Note: in the following, each original reviewer comment is quoted in *italics*, and our response given in plain text.

**Response to Reviewer 1 comments:**

*The main criticism on the paper relates to the model description. The authors present The Grain Hill model in Section 2 ('Model description'). This model builds on the previously introduced (slightly modified) CTS Lattice Grain model (Tucker et al., 2016) which is the main dynamical component of Grain Hill determining the movement of the grains on the hillslope. Although the authors have made some efforts to briefly describe this model component, I have found it insufficient. I am not claiming that the authors should provide all the details (again) of this model, since that description was the object of a*

[Figure]

*previous publication, but given that the scope of the paper is introducing a new model for hillslope evolution (e.g. as indicated by the title), this paper should be sufficiently self-contained. What I mean is that the reader should have a good understanding of the functioning of the model (rules and dynamics) without the need of going back and forth between this manuscript and Tucker et al. (2016), which I have found myself doing quite often. Obviously, the technical implementation of the model rules (e.g., smart algorithms to rank times or how to take advantage of the Landlab's grid architecture) can be omitted, since they are well described in the previous publication, and it is not detrimental to conveying the essence of the model.*

This is a fair point, and we appreciate the value of having this paper be accessible on its own without forcing a reader to delve through the prior publication on CellLabCTS. In the revised manuscript, we have added text to the model description section that provides a more thorough explanation of the rules and how they function. Specific examples of changes made are given below in response to more detailed comments along these lines.

*My understanding of the grain motion for the most basic grain dynamics of the model is as follows: (1) Disturbance is described as a Poisson process, assigning a state transition at a given time for every surface grain (random variable taken from an exponential distribution with parameter d), which forms a queue sorted by the transition times. (2) The "soonest transition" (change in the state of the corresponding grain) in the queue is executed, which may trigger future state transitions of neighboring grains. (3) The gravity and inelastic collision rules (Figure 4 and 5) would dictate the subsequent transitions and movements. I assume that the two latter processes (driven by gravity and collision) operate at smaller time scales. Is this so?*

Yes, that is correct. We have added a new paragraph to section 2.1 to explain that each of the various transitions is represented as a Poisson process with a given rate constant. This paragraph briefly describes the "scheduling" process by which transitions

are evaluated. We also added a short paragraph at the end of the section that notes the expectation of two distinct time scales.

*What is the time scale of the gravitational transitions illustrated in figure 5? is that time scale dictated by an additional parameter/rate g (not listed in this manuscript)?*

Yes, there is indeed an additional gravitational rate parameter. This parameter is the inverse of a gravitational settling time scale. Let's call this Tg. It represents the time it would take an initially stationary object to fall a distance of one cell (delta), assuming no fluid drag. Thus, g depends on the planet (we assume earth!) and on delta. To be exact, the time constant Tg = sqrt(2 delta/g). For most of the simulations shown in the paper, the cell size is assumed to be 0.1 m, and the corresponding Tg is 0.14 s, and the settling transition rate parameter is $\sim$7 s$^{-1}$. Because the time units in Grain Hill are taken to be years, the settling rate parameter is actually entered in units of inverse years: $1/Tg \sim 2.2 \times 10^{-7}y^{-1}$. The basis for the gravitational rate constant is now explained in section 2.1.

*is g a constant rate, and therefore all the grains transition exactly at time 1/g after being updated to their current state? or is g the rate that characterizes an exponentially distributed random variable $\tau$, and each grain transitions after a (different) lag $\tau$?*

Excellent question. It is the second: gravitational settling is treated as a Poisson process, just like all other transitions in the model. This is now clarified in section 2.1. Admittedly, the stochastic treatment of grain settling raises the distinct possibility of having a grain hover in mid-air for a while before suddenly plunging downward (this is now acknowledged in the text). For our purposes, that's ok. What matters is that the grain-motion dynamics are fast relative to the "geomorphic" processes. This conclusion is backed up by informal sensitivity analysis, which shows that the choice of settling rate doesn't matter much as long as it is much shorter than the characteristic time scale for weathering, disturbance, and uplift relative to baselevel.
*o What is the time scale at which the inelastic collisions and movements illustrated in figure 4 take place? Same than the previous time scale,1/g? constant lag or stochastic random variable?*

Again, great question. Motion and collision transitions are both treated as stochastic. The model is built around the gravitational acceleration time scale. For the Grain Hill implementation, because collisions are assumed to be inelastic, the rate constant is the same as the gravitational rate constant. This is now explained in the revised version of the manuscript.

*My understanding is that when a grain (or aggregate) moves to occupy an empty cell (fluid), it keeps the same state of movement (arrow direction). Is that correct?*

Yes, that is correct. We have added a short paragraph explaining how motion works, including the fact that motion direction remains unchanged. As noted below, we also replaced one of the figures with a version that illustrates several different kinds of transition, including motion.

*If yes, can this give rise to grains that temporally float in the fluid (grain surrounded by six fluid cells)? (I think it depends on the previous questions about the characteristic time scales of gravitational and movement/collision transitions).*

Absolutely. This represents a grain in mid-trajectory. An example of this now appears in the new replacement figure.

*How is the model initialized? For example, what is the geometry (flat?)? Do the lateral edges of the grid play a main role on the early evolution of the hillslope?*

The model is initialized as mostly fluid, with regolith cells in the lowest two rows of the model domain (not including the left and right sides). The lower left and lower right cells are assigned to be rock, which represents the baselevel (and incidentally helps keep a consistent color scheme among different model configurations, because the rock state

is always present). This is now explained in a the subsection "Initial and Boundary Conditions."

*How does the grid type (squares vs. hexagons) affect the results? It is stated in page 7 (lines 12- 13) that the gravitational rule sets effectively the angle of repose to 30 degrees. Does the angle of repose strongly relate to the chosen hexagonal grid?*

It is hard to answer this question definitively, because we have not tried running the model with a square lattice. Clearly, however, it is the combination of the hex grid and the down-and-to-the-side gravitational rule that imposes the 30 degree angle of repose. As the text notes, this restriction to 30 degrees is a limitation of the model. It could potentially be overcome by generalizing the transition rules to allow for a finite probability that a particular transition type will not take place at a particular location— and, in this particular example, that a spontaneous "sideways fall" will only occur with a probability P; otherwise, the material at that location is considered stable. One would then have two random variables associated with each transition type: one representing the probability that the transition will indeed occur at some future time, and (if that condition applies) the time in the future at which the transition will take place.

We should add that the hex geometry was chosen in part because it allows more axes of symmetry (3) than a square lattice (2). The alignment of the hex grid, with one axis vertical, was also deliberately chosen to allow for direct vertical fall of particles.

*I would really appreciate as a reader to have access to some videos (as supplemental information) of the evolution of the model, including the evolution of the hillslope as it approaches its steady state.*

Good idea. We have created five videos that will be added to the supplement.

*Figure 3 is not very straightforward. Maybe it would be good to highlight the cell-pair affected by the transition/movement. With the current caption and text, it is confusing to see several arrows, but only one cell pair changing (this takes me back to my first*

*questions about time scales). Also, I guess that this picture is not illustrative of the The Grain Hill model (otherwise some grains would be floating on the air/fluid). Maybe it would be more interesting to provide an example relevant to hillslopes.*

We have replaced this with a figure that shows a sequence of transition events, and added explanatory text to the caption. This figure is now referred to in a couple of additional places in the text.

*The authors illustrated in Figure 19 that the model is capable of reproducing realistic hillslope forms with parameters estimated from field data. However, it is not straightforward to evaluate the model performance in the absence of a direct comparison with the observations. I suggest that the authors can either (a) overlap Figure 19a,b with the profiles shown in Figure 2a,b to show that qualitatively the model is able to reproduce the form and height of those landforms, or (b) provide a quantification of the fit of the observed profile with the model results.*

Good point. We have added the observed profiles to Figure 19.

*Regarding the determination of the parameters, it may also be helpful if the authors can comment on their assumption of δ=1 m, as they suggested that the soil depths typically range between 0.2 and 1.2 m which is almost an order of magnitude difference for the variation. Are the results sensitive to this parameter?*

We added a short paragraph summarizing the result of a sensitivity test on the Yucaipa example, which shows the one obtains the same shape and relief when delta is changed and the other parameters rescaled accordingly.

*Figure 7 looks disproportionally big in comparison with other figures (e.g. Figure 4 or 5).*

Figure 7 is meant to be a smaller, single-column figure. It is now re-scaled in the PDF file.

*Please label the different panels of Figure 8 (e.g. a,b,...) to make easier their reference in the text.*

We actually prefer not to do this, because the key text references to parts of this figure refer deliberately to geometric groupings: upper left, middle diagonal, lower right. If we labeled them with letters, readers would have to take time to work out which ones are a,b,d or f,h,i. The geometric references shouldn't be any more complicated (in fact, probably more intuitive), and it avoids cluttering the figure.

*The authors state that Figure 9 shows the results of 125 model runs. It is not clear what those different runs are. From the figure, I can see 5x9 different combinations of $\lambda$ and d so, do you run some cases twice and some case three times?*

Added text here to explain that the 125 runs represent a 5x5x5 experimental grid, in which each grid point represents a particular combination of the three parameters d, $\tau$, and $\lambda$.

*Figure 19 caption: "State state models" –> "Steady state models".*

Fixed, thanks

*Given the scope of the paper and the number of figures, I am not sure if Figure 6 adds much to the explanation of the model.*

We think it is important to establish that the Lattice Grain model captures the essence of loose granular materials under gravity. We could simply refer to the original paper, but the reviewer noted above that it is easier if this piece can stand alone without the need to refer back to Tucker et al. (2016), so we prefer to keep this figure.

*Please be consistent in terminology (e.g. mean gradient vs. slope gradient)*

Thanks, fixed a few of these. However, there are places where the distinction between (local) slope gradient and mean slope gradient is important, so we've kept those as is.

*The authors emphasize the simplicity of the model by pointing out the reduced number of parameters (3 in its simplest version). However, this kind of assessment is a little tricky when we refer to CA models, where the type and number of rules play a significant role in evaluating the simplicity of the model.*

This is a fair point. The motion/gravitation rate can be thought of as a parameter, though as noted above, we think the model is insensitive to its value provided there is a clear gap between the "fast" and "slow" processes. Regarding the rules themselves, we regard these as being equivalent to terms in a traditional differential-equation model: they describe the physics rather than the rates (or length scales, etc.).

*For me it is a little contradictory using "continuous-time" and "Cellular Automaton" as descriptors of a model. Cellular automata were originally defined as discrete models both in time and space, as well as the state variable of each grid cell. Models with discrete space and state variables, but continuous time, sometimes are called Boolean Delay Equations (Figure 2 in Zaliapin et al. 2003).*

We appreciate hearing about this classification scheme. It seems, however, that there is not a consistent nomenclature. For example, Narteau and colleagues refer to their stochastic-in-time approach as a cellular automata, where Zaliapin would call it a Boolean Delay Equation. We can't be consistent with both at the same time, and in our earlier paper we called the approach a cellular automaton, so it seems best to stick with that terminology. However, we have added a nod to this with a sentence along the lines of: "A CTS model can be viewed as a type of Boolean Delay Equation (Ghil et al., 2008), though the number of possible states is not necessarily limited to just two."

**Response to Reviewer 2 comments:**

*This manuscript describes a new method to simulate hillslopes based on a lattice-based cellular automaton framework. This work is highly innovative, clever, and wonderfully presented and I recommend publication given some minor considerations and*

*clarifications that I'll detail below.*

*First, though, I'll note that this manuscript is situated on the cusp of invigorated efforts to revisit how our community does the accounting of mass on hillslopes. The continuum perspective that pervades so much of geophysical theory and simulation is being revisited to account for stochastic, nonlocal, and other effects and the approach adopted here complements forward-looking work by Furbish, Ganti, and others. While cellular automaton schemes have been around for some time, I am not aware of studies that are so carefully crafted as this one as to allow for comparison and calibration with commonly used continuum parameters. This manuscript offers a clear roadmap for applying CA schemes to well-trod field sites and characteristic landforms.*

*Second, this manuscript is innovative not only for the algorithmic advances but the interpretations that emerge are also highly compelling. In particular, the ability of the model to account for disturbance processes as the source of both soil transport and production is conceptually appealing and a long-overdue advance. The continuum requirement of breaking up these processes has always been unsatisfying and this effort is to my knowledge the first to provide a unifying treatment. Also, the model's ability to account for retreating cliff-rampart and rocky landforms is highly compelling. The authors neatly applied their toolkit to two field sites as a means to independently confirm their scaling arguments for the model parameters.*

*Third, (I'm still in gushing mode), I'll note that the lattice grain model presented here is built on parsimony from the parameterization standpoint, which enables the authors to efficiently study a wide range of model outcomes.*

Thanks for these comments! Indeed the approach has been inspired by advances in several directions, including recent efforts to understand what is sometimes called "stochastic transport" (as explored by the authors listed above, among others), as well as by our own frustrating experiences in trying to apply continuum theory to systems

with complicated geometries and boundary conditions.

*Below, I detail several comments for the authors to consider.*

*1) While the authors claim to invoke minimal parameters in defining their simulations, it seems that the choice of hexagon elements (which conveniently pack purely) as well as the myriad rules for different geometries constitute choices that impact the model outputs. For example, the hexagon packing results in a 30 degree angle of repose slope due to the geometry of the hexagon contacts as well as the gravity rules. This is an arbitrary choice that's convenient for geometry and computational purposes. It turns out that 30 degrees is a fairly common friction angle for smooth cohesion grain piles, so this is appealing. That said, what prevents the model from being performed with octagons? In that case, the packing isn't as efficient and voids are required but it seems defensible nonetheless. Such a geometry would result in a 45 deg friction angle if the same rules were applied. Other element shapes and packing schemes are possible, in fact, a mind-boggling array could be tested. How does the choice of hexagons affect the results? Beyond setting the angle of repose, how do different packing schemes affect the rules for state changes (see below)?*

It's a great question, and one that we have addressed to some extent above, in response to a similar comment from Reviewer 1. Simply put, we haven't studied other configurations. The most practical choices are squares or hexagons, and we picked hexagons because they offer an extra degree of geometric freedom. But it would be useful if there were a means to allow angle-of-repose behavior to reflect something other than grid geometry. To address this issue, the Discussion section notes the possibility of relaxing the 30 degree constraint with a "sticking rule". We added a comment along the lines of: "Such an enhancement might also make it possible for models with alternative lattice geometries to manifest the same dynamics, thereby de-coupling the basic model framework from the geometry of the lattice on which it is implemented." We hope this captures the spirit of the reviewer's point that the lattice is in a sense part

of the model.

**ESurfD**

Interactive
comment

*2) The rules of state changes are nicely laid out with an example orientation in figure
4, although it seems that some possible states are not represented and it would be
nice if the figure was comprehensive given that no standards are offered as to why
certain motion orientations result in motion and others do not. Can the authors write
criteria for the outcomes of the inelastic collisions shown in Figure 4? In particular,
the two particles that come together (far right) generate split 50/50 outcomes, why
doesn't the same type of outcome emerge with the far left shear-like interaction? I
suspect that some of these details don't matter terribly much but it'd be interesting to
better understand which do matter and thus account for the behavior observed in the
simulations. Is it possible to do an accounting of the state transformations and where
they occur in order to determine which interactions dominate in the physical space of
the model? Such an exercise is probably beyond the scope of this paper but it'd be
very interesting to better understand the impact of the interaction choices listed here.*

Indeed, the collision rules are somewhat arbitrary, and one could reasonably justify
different rules from the ones illustrated here. For example, more of the collisions could
lead both particles to halt (instead of just one), or conversely. We have not done com-
prehensive testing of these rules (other than to demonstrate that they lead to qualitative
behavior that seems to mimic real sandpiles). There is certainly scope for a compre-
hensive sensitivity analysis of the collision rules, though not in this particular paper.
We have added text in the model description section to the effect that: "The particular
choices for frictional interaction are motivated simply by the geometry of the problem.
They are non-unique in the sense that one could imagine reasonable alternatives to
the rules illustrated in Figure **??**; however, the details of frictional interactions have little
influence on the outcomes of the Grain Hill model."

*3) I'm unclear on how the state transition occurs during a given time step. Given that
state rules are only defined pair-wise, how are the transitions generated given that each*

*element has 6 neighbors? How is precedence or primacy established? I'm certainly missing some aspect of how these calculations are performed, so perhaps a bit more explanation on this point is possible? Do the pair-state calculations actually occur in sequence? Or does the state transition happen simultaneously? If so, how are the particular pairs chosen? Or is every combination assessed?*

As noted above in response to a comment by Reviewer 1, the model does not use time steps, but rather iterates through a series of asynchronous transition events. Text has now been added to the manuscript to explain this more clearly.

*4) While the text addresses previous some work on soil-mantled slopes and transport models, the description of rocky slopes and cliff-rampart settings is lacking in scholarship and context. There's a rich literature about cliff retreat in which conceptual models posed are likely consistent with the results generated here. Many workers, including Davis, King, Oberlander, Twidale, Ollier, and others have contemplated these landforms with varying process models in mind, including and notably overland flow. Onlya very recent paper is cited here. The fact that the lattice grain model can generate iconic landforms without directly invoking overland flow is compelling and worth mention and/or context.*

It's absolutely true that a lot has been written about rocky slopes in the "classic" literature. The problem is that there's so much that it would be impossible to summarize without detracting from the flow of the paper. To some extent, that literature is encapsulated in textbooks, but it doesn't feel right to quote a text. We chose the Howard and Selby paper is a fairly comprehensive review. In order to indicate that this is but one of many, we have now decorated that reference with "and references therein." We realize that this is not an ideal solution, but given that the main aim of this paper is to introduce the Grain Hill model as a concept, it seems like a reasonable shortcut. We envision a future paper that focuses less on the technique and more on the implications for rock slopes, where it would be appropriate to delve more deeply into the classic literature

on such slopes.

*5) The emergence of steady state rocky landforms for baselevel lowering rates that exceed the maximum soil production rate is perhaps the most attention-grabbing result of this model. The authors nicely describe how this results from slope and roughness, which is intuitive upon reflection. Given the importance of surface area for the total weathering flux, do any characteristic wavelengths emerge along the interface? is the roughness isolated to the element-scale or is it superimposed on broader wavelength fluctuations? It would be interesting to see if complexity (1/f noise or otherwise) of the surface emerges due to the need for weathering rates to keep pace. The possibility for feedbacks seems like fertile territory.*

Great question. We haven't noticed emergence of characteristic wavelengths above the cell scale, but agree that this is fertile territory for future study.

*6) The connection between disturbance and weathering through exposure is very appealing and nicely stated. In this sense, the creation of porosity and the evolution of that porosity in the regolith is a major conceptual advance. That said, it isn't not clear to me in the text that porosity can be essentially advected to depth given a particular sequence of phase state changes. Is that the case? I'm not sure how else porosity goes downward, but however it does, it's worth clarification b/c this sets the weathering rate along the bedrock interface. This goes back to the choices about pair-wise interactions. Would other choices for the interaction rules result in more/less porosity advection?*

What a neat idea! It's a limitation of the model in its present form that there is no "explicit" porosity. Under the current rules, if a "pore cell" were to appear, it would rapidly be closed by collapse of the overlying particle. However, if one enhanced the rules with a "sticking probability", then conceivably one could have perfectly stable pores. Then one could actually begin to explore porosity advection. There might also need to be other, deeper changes, such as the use of larger cell neighborhoods (e.g.,

6 cells instead of 2) in order to express cohesive bonding. We haven't tried to address this in the revised text, but would be eager to discuss these ideas further with the reviewer.

*minor comments:*

*1) abstract: line 6: the concept of disturbances is part/parcel of this contribution and may merit mention in the abstract. Otherwise, the large block comment is unclear and is probably secondary in importance.*

Good point – added text along the lines of: "Cells representing near-surface soil material undergo stochastic disturbance events, in which initially stationary material is put into motion and is statistically more likely to move downhill than uphill before coming to rest again. Net downslope transport emerges from the greater likelihood for disturbed material to move downhill than to move uphill. Cells representing rock undergo stochastic weathering events in which the rock is converted into regolith. "

*2) pg 2, line 38ish: is it worth mentioning that the lattices don't actually move but rather than the state evolves? it took me awhile to come around to this realization and it could be stated earlier in my opinion.*

Great suggestion; now added a sentence to note this fact.

*3) pg 4, line 20-25: the concept of state is employed here before it's explained, I recommend moving some of the text from below (30-33 or so) upwards to clarify.*

Added a bit of text here to indicate that a state is represented by an integer value at each cell.

*4) pg 7, line 22: given the porosity, would it be possible to include water and a weathering function in the future?*

That would probably require an additional set of rules. For example, it would probably
be necessary to add a particle state that represents pore water or saturated soil. But in principle the framework could accommodate this sort of thing.

*5) pg 10, line 16: given comments above about hexagon and friction angle, isn't there really more than four parameters? I'm not sure how the counting works, but my sense is that the hillslope relief should depend on the angle of repose which results from hexagon worldview.*

Fair enough, though we actually view the hex geometry as part of the "governing equation" rather than a parameter. Nonetheless, we nod to this issue in the Discussion section in a new sentence noting the desirability of de-coupling the model from the lattice geometry.

*6) pg 13, figure 8: would it be possible to fit the standard paraboblic (and or nonlinear) analytical curves to these experimental profiles? that would be another way to determine the parameter scaling.*

Actually, that's almost what we've done: we're fitting the average height of the standard parabolic curve to the average height in the experimental profiles. Text has now been added to point this out.

*7) pg 14, eqn 13: I'm dim, but why is 3 on the denominator rather than 2? My apologies but whenever I've done this integration I end up with 2 on the bottom...please help.*

It basically comes from integrating twice (once to go from slope to elevation profile, and a second time to get the average elevation). The derivation is:

$$\int_0^\eta d\eta = -(E/D_e) \int_{L_h}^x x\,dx \rightarrow \tag{1}$$

$$\eta - 0 = -(E/D_e)(x^2/2 - L_h^2/2) \rightarrow \tag{2}$$

$$\eta = (E/2D_e)(L_h^2 - x^2) \tag{3}$$

That's the parabolic profile. Then integrate to get the average height:

$$\bar{\eta} = (1/L_h)(E/2D_e) \left[ \int_0^{L_h} L_h^2 dx - \int_0^{L_h} x^2 dx \right] \rightarrow \tag{4}$$

$$\bar{\eta} = (1/L_h)(E/2D_e)[L_h^3 - L_h^3/3] \rightarrow \tag{5}$$

$$\bar{\eta} = (1/L_h)(E/2D_e)(2/3)L_h^3 \rightarrow \tag{6}$$

$$\bar{\eta} = (E/3D_e)L_h^2 \tag{7}$$

*8) pg 19, figure 13: it's difficult to see the various symbols here.*

Yes, that's true—many are over-printed because they fall right on the 1:1 line. Because there's no obvious way to correct this on the plot itself, we have added a line of text pointing out to the reader that some of the lower-quintile points are obscured by being over-plotted right on the 1:1 line.

*9) pg 20, line 4: isn't the creation of porosity and it's downward advection akin to a depth-dependent process? the text here seems restrictive.*

Porosity is only being created indirectly, through the process of rock-to-regolith conversion (as opposed to being explicitly represented by fluid cells in the subsurface). Under the rules presented here, rock-to-regolith conversion only happens where a rock cell meets a fluid cell, which is why we say here that the model doesn't capture weathering processes that extend deeper than one cell at any given moment. However, the caveat "present configuration" is meant to signal that one COULD develop more sophisticated rules that would relax this limitation. For example, you might introduce a "saturated regolith" condition that could move downward (representing infiltration) or perhaps upward (representing capillary suction). One could then introduce a transition wherein the pairing of rock with saturated regolith transitions to two regolith (wet or dry) cells, representing subsurface chemical weathering.

*10) 7) The diffusivity values for Gabilan Mesa seem quite high and I wonder if this results from K calibration using low-curvature hilltops that are not reflective of integrated erosion in that site.*

The value is indeed on the high end, though not off the charts (McKean et al.'s estimate from the Black Diamond Reserve in CA was 0.036 m2/yr). We estimated a profile-averaged effective transport coefficient. The Gabilan profiles are not perfectly parabolic and would be best fit by a slightly nonlinear transport rule. The slightly high D reflects this. A transport coefficient obtained by only fitting the constant curvature hilltops is indeed lower than what we report for the profile-averaged value.